# Repeated activation of preoptic area recipient neurons in posterior paraventricular nucleus mediates chronic heat-induced negative emotional valence and hyperarousal states

Zhiping Cao[1], Wing-Ho Yung[2]*, Ya Ke[1]*

[1]School of Biomedical Sciences, Faculty of Medicine, The Chinese University of Hong Kong, Hong Kong, China; [2]Department of Neuroscience, College of Biomedicine, City University of Hong Kong, Hong Kong, China

*For correspondence:
whyung@cityu.edu.hk (W-H);
yake@cuhk.edu.hk (YK)

**Competing interest:** The authors declare that no competing interests exist.

## eLife Assessment

This **important** study identifies one way in which episodic heat exposure can result in negative changes in motivated and affective behaviors. This work positively expands the field of thermoregulation. The data were collected using a myriad of next-generation approaches, including extensive behavior testing, thermal monitoring, electrophysiology, circuit mapping, and manipulations. There is **convincing** evidence that neurons of the paraventricular thalamus change plastically over three weeks of episodic heat stimulation this affects behavioral outputs such as social interactions and anxiety-related behavior. Conclusions regarding the specificity of the POA-pPVT pathway compared to other inputs to the PVT in the control of observed effects would benefit from further validation. The study will be of interest to behavioral neuroscientists, climate/environmental biologists, and pre-clinical neuropsychiatrists.

**Abstract** Mental and behavioral disorders are associated with extended period of hot weather as found in heatwaves, but the underlying neural circuit mechanism remains poorly known. The posterior paraventricular thalamus (pPVT) is a hub for emotional processing and receives inputs from the hypothalamic preoptic area (POA), the well-recognized thermoregulation center. The present study was designed to explore whether chronic heat exposure leads to aberrant activities in POA recipient pPVT neurons and subsequent changes in emotional states. By devising an air heating paradigm mimicking the condition of heatwaves and utilizing emotion-related behavioral tests, viral tract tracing, in vivo calcium recordings, optogenetic manipulations, and electrophysiological recordings, we found that chronic heat exposure for 3 weeks led to negative emotional valence and hyperarousal states in mice. The pPVT neurons receive monosynaptic excitatory and inhibitory innervations from the POA. These neurons exhibited a persistent increase in neural activity following chronic heat exposure, which was essential for chronic heat-induced emotional changes. Notably, these neurons were also prone to display stronger neuronal activities associated with anxiety responses to stressful situations. Furthermore, we observed saturated neuroplasticity in the POA-pPVT excitatory pathway after chronic heat exposure that occluded further potentiation. Taken together, long-term aberration in the POA to pPVT pathway offers a neurobiological mechanism of emotional and behavioral changes seen in extended periods of hot weather like heatwaves.

## Introduction

Against a backdrop of global warming, heatwaves that are characterized by abnormally hot weather for extended periods have become more frequent and intense. This is highlighted by that the July 2023 was the hottest month ever recorded by human (*Tollefson, 2023*). Traditional views support that repetitive heat exposure and the related heat acclimation have some beneficial effects in allowing the animals and humans to gain heat tolerance, such as strengthening the cardiovascular system, reducing energy metabolism, and weight (*Murray et al., 2022*; *Podstawski et al., 2021b*; *Périard et al., 2016*; *Fausnacht et al., 2021*). However, many studies have also established a correlation between heatwaves and impaired physical health leading to increased mortality (*Barriopedro et al., 2011*; *García-Herrera et al., 2010*; *Mitchell et al., 2016*; *Mora et al., 2017*; *Robine et al., 2008*). In addition, elevated temperatures could impact mental health by triggering feelings of anger, stress, aggression, and depression (*Beecher et al., 2016*). In fact, epidemiological research conducted across various regions also extensively demonstrated a positive association between chronic heat exposure and higher hospital admissions for mental and behavioral disorders (*Amr and Volpe, 2012*; *Basu et al., 2018*; *Dominiak et al., 2015*; *Hansen et al., 2008*; *Lee et al., 2002*; *Mulder et al., 1990*; *Trang et al., 2015*; *Whitney et al., 1999*). Nevertheless, our understanding of how the brain regulates emotional changes following prolonged heat exposure remains limited.

Current understanding about heat-induced emotional changes mainly focuses on the impact of heat stress on the hypothalamic-pituitary-adrenal (HPA) axis, which leads to abnormal plasma concentrations of the hormone cortisol and neurotransmitters such as 5-hydroxytryptamine, noradrenaline, and adrenaline, all of which play a significant role in modulating emotional states (*McMorris et al., 2006*; *Wang et al., 2015*). However, how heat affects emotions from the perspective of neural circuits has not been substantiated. Moreover, the relationship between chronic heat exposure and emotional changes is likely to be more complex. Multifaceted factors, such as the levels of physiological stress (*Sampath et al., 2023*), changes of neuroendocrine system (*Podstawski et al., 2021a*), sleep disturbances (*Altena et al., 2023*), adaptation process (*Oppermann et al., 2021*), even the age, and individual health status (*Kenny et al., 2010*; *Malmquist et al., 2022*), could interact in various ways to directly and indirectly contribute to emotional changes in response to chronic heat exposure. Thus, gaining insights into the circuitry mechanisms behind emotional changes caused by chronic heat exposure is crucial for a more comprehensive understanding of the neurobiological foundations of emotional regulation.

Emerging evidence has suggested that the paraventricular thalamus (PVT) serves as an important integrative node that detects aversive sensory and homeostatic challenges and regulates emotional responses and adaptive behaviors. As such, PVT neurons receive inputs from diverse nuclei in the hypothalamus, midbrain, and hindbrain, and has been implicated in a variety of behavioral responses, including arousal (*Ren et al., 2018*), pain (*Jurik et al., 2015*), anxiety (*Heilbronner et al., 2004*; *Li et al., 2010*; *Spencer et al., 2004*), fear, and fear memory (*Do Monte et al., 2015*; *Penzo and Gao, 2021*; *Penzo et al., 2015*). Therefore, PVT is now considered as an essential component of the emotional processing system in the brain (*Penzo and Gao, 2021*).

Among the various inputs from different brain regions to the PVT, the hypothalamus preoptic area (POA) is a well-known thermoregulatory center of the brain (*Yu et al., 2016*; *Zhang et al., 2011*; *Zhao et al., 2017*). Anatomically, the posterior PVT (pPVT) has been reported to receive projections from POA warm-sensitive neurons (*Tan et al., 2016*). Here, we hypothesize that the direct projection from POA to pPVT contributes to chronic heat-induced emotional disturbances. Based on the murine model, we dissected the synaptic connection from POA to pPVT, tracked its activities, and interrogated its involvement in behavioral responses of the animals under acute and chronic heat exposure paradigms. Our results uncovered long-term aberration in the activity of this pathway and pPVT neurons mediating chronic heat-induced changes in emotional and hyperarousal states, providing a neurobiological basis of how chronic heat affects mental states.

## Results

### Chronic heat produces negative emotional valence and hyperarousal states but not depression-like behaviors in mice

C57BL/6 mice were exposed to chronic heat by putting them into a pre-heated (38±2°C) chamber daily for 21 days with free access of food and water (*Figure 1A*, details in Materials and methods). In our preliminary experiments, various durations of daily heat exposure were tested. We settled to 90 min due to its robustness in inducing emotional state-related behavioral changes without causing the collapse of the animals. Throughout the entire process of chronic heat exposure, we monitored the physiological states of mice daily. We did not observe abnormal changes in body temperature the day after chronic heat exposure (*Figure 1—figure supplement 1A and B*) but affected normal gain in body weight (*Figure 1—figure supplement 1C and D*) and reduced food consumption (*Figure 1—figure supplement 1E and F*), compared with control mice. On the other hand, mice subjected to chronic heat displayed a higher prevalence of stress responses in various behavioral tests. Specifically, in the elevated plus maze test, chronic heat-exposed mice spent less time in the open arms (*Figure 1B and C*), indicating increased stress levels. Moreover, in the three-chamber test, these mice displayed reduced interests in exploring the unfamiliar male mouse compared to an inanimate object (*Figure 1D and E*), suggesting decreased sociability. Similarly, during the female encounter test, these mice showed less engagement with the unfamiliar female mouse compared with the unfamiliar male mouse (*Figure 1F and G*), indicating decreased innate motivation. Interestingly, the chronic heat-exposed mice displayed decreased latencies for the first attack but increased attack durations in the resident-intruder test (*Figure 1H and I*), indicating elevated aggression levels.

We also observed that after chronic heat exposure, the mice displayed heightened alertness, suggesting a state of hyperarousal. To confirm this, we quantified the acoustic startle response (ASR) of the animals by digital video capture of their rapid movements (*Pantoni et al., 2020*). Chronic heat-exposed mice displayed exaggerated ASR characterized by increased body fluctuations within 200 ms of the delivery of a 105 dB sound stimulus (*Figure 1J–M* and *Figure 1—video 1*). However, these mice did not exhibit clear hyperlocomotion during the three-chamber test, the female encounter test, or the open-field test (*Figure 1—figure supplement 1G–I*), suggesting increased agitation in mice. Collectively, our data demonstrate that chronic heat exposure triggers increased stress responses as well as increased arousal in mice.

The fact that the observed changes in emotional and arousal states are the results of long-term rather than short-term heat exposure is supported by the lack of effect measured 1 day following one-time acute heat exposure for 90 min (*Figure 1—figure supplement 2*). Furthermore, we did not observe signs of depression in chronic heat-exposed mice, as there were no significant increases in immobile time during forced swimming (*Figure 1N*) and the tail suspension tests (*Figure 1O*), no obvious decrease in sucrose intake (*Figure 1P*), and no sign of decreased exploration as there was no change in the time spent by these animals in the center zone of the open field (*Figure 1Q*).

### Involvement of the hypothalamic POA to pPVT projection

To explore the involvement of POA and pPVT in chronic heat exposure-induced behavioral changes, we first studied the effect of heat treatment per se on their activities via c-Fos staining. We found that acute heat treatment (for 90 mins) significantly increased c-Fos expressions of POA and pPVT neurons (*Figure 2—figure supplement 1A–C*), supporting not only the notion that the POA serves as a thermoregulation center (*Morrison and Nakamura, 2011*) but also the recruitment of pPVT following heat treatment. Next, to dissect the anatomical relationship between POA and pPVT especially in the context of heat exposure, we injected viruses (AAV1-hSyn-cre, AAV-FLEX-TVA-mCherry, AAV-FLEX-RG, and subsequently SAD△G (EnvA) virus) of a monosynaptic retrograde tracing rabies virus system (details in Materials and methods) into pPVT, which resulted in clear expression of retrogradely labeled neurons within POA, its upstream area (*Figure 2A–C*). Moreover, a significant overlap (84 ± 5%, 3 mice) was observed between the retrogradely labeled and c-Fos expressed neurons in POA following heat exposure (*Figure 2D*). Furthermore, almost all the POA recipient pPVT neurons identified by a Cre-dependent anterograde labeling strategy (93 ± 2%, 3 mice) were strongly activated by heat exposure (*Figure 2—figure supplement 2*). Together, our results indicate that most of the pPVT-projecting POA neurons were neurons that respond to heat treatment which would then recruit their downstream neurons in pPVT by exerting a net excitatory influence.

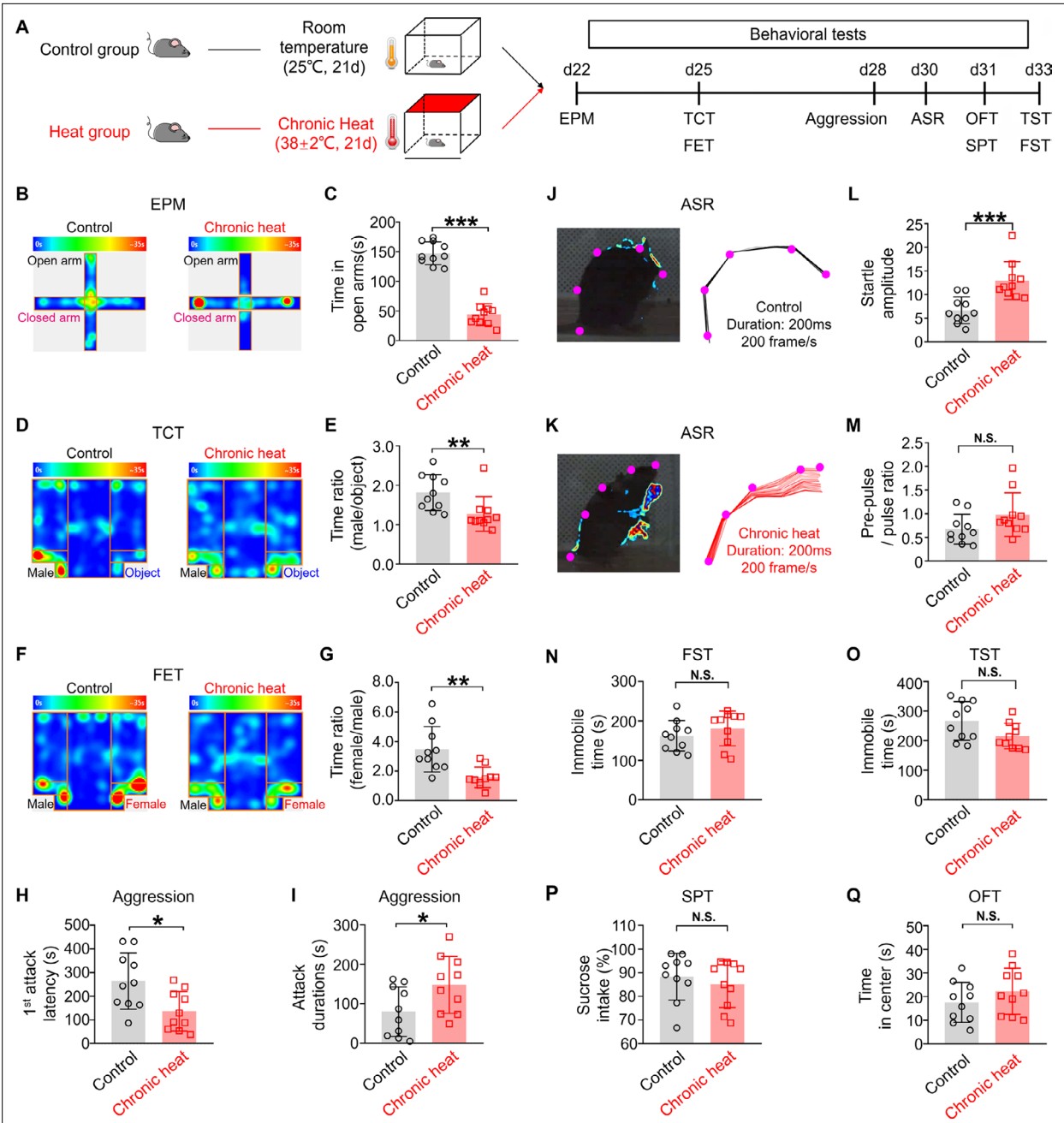

**Figure 1.** Chronic heat exposure produces negative emotional valence and hyperarousal states but not depression-like behaviors. (**A**) Experimental schematics. Mice (n=10 in each group) were divided into Control and Heat groups and conducted with chronic exposure to room temperature and heat conditions, respectively, followed by behavioral tests. (**B, C**) The heatmaps of representative track tracing in elevated plus maze (EPM) test and the time spent in the open arms of EPM (Mann-Whitney unpaired two-tailed U test; U=0, ***p<0.001). (**D, E**) The representative heatmaps of in three-chamber test (TCT) and the interaction time with an unfamiliar male mouse relative to an inanimate object in TCT (Mann-Whitney unpaired two-tailed U test; U=13, ***p=0.0039). (**F, G**) The heatmap of representative tracking trace examples in female encounter test (FET) and the time surrounding the unfamiliar female mouse compared to an unfamiliar male mouse in FET (Mann-Whitney unpaired two-tailed U test; U=10, **p=0.0015). (**H, I**) The first-time attack latency (Mann-Whitney unpaired two-tailed U test; U=21, *p=0.0288) and the attack durations (Mann-Whitney unpaired two-tailed U test; U=23, *p=0.0433) in the aggression test. (**J, K**) Visualized and representative acoustic startle response (ASR) example (left panel) and corresponding labeled body parts' skeletons (shown as purple dots) (right panel) in ASR test from control and chronic heat group.(**L, M**) The startle amplitude (Mann-Whitney unpaired two-tailed U test; U=6, ***p=0.0003) and the pre-pulse/pulse ratio (Mann-Whitney unpaired two-tailed U test; U=26, p=0.0753) in ASR test. (**N–Q**) The immobile time in the forced swim test (FST) (Mann-Whitney unpaired two-tailed U test; U=35, p=0.2799) and in the tail suspension test (TST) (Mann-Whitney unpaired two-tailed U test; U=26, p=0.0753). The percentage of sucrose intake in the sucrose preference test (SPT) (Mann-Whitney

*Figure 1 continued on next page*

*Figure 1 continued*

unpaired two-tailed U test; U=49.5, p=0.4887) and the time spent in the center of the open field test (OFT) (Mann-Whitney unpaired two-tailed U test; U=35.5, p=0.2888). *p<0.05, **p<0.01, ***p<0.001, NS: not significant.

The online version of this article includes the following video and figure supplement(s) for figure 1:

**Figure supplement 1.** The effect of chronic heat exposure on physiological states of mice and their motion activity during behavioral tests.

**Figure supplement 2.** Mice did not exhibit obvious changes of emotional valence and arousal states the day after acute heat exposure.

**Figure 1—video 1.** Representative video showed the delivery of a 105 dB sound stimulus within 200 ms evoked an obvious body fluctuation in the chronic heat-exposed mouse.

https://elifesciences.org/articles/101302/figures#fig1video1

To functionally characterize the synaptic connection between POA and pPVT, we administered AAV9-hSyn-ChR2-mCherry into the POA for optogenetic manipulation of its neurons and their terminals in pPVT. By patch-clamp recordings in brain slices prepared from these animals, we first confirmed robust firing of ChR2-mCherry-expressing neurons in POA upon light stimulation at various frequencies (*Figure 2E*). When we applied 473 nm light stimulation to POA terminals in pPVT, in some neurons, a clear inhibitory outward postsynaptic current was recorded in pPVT neurons held at +10 mV, which was sensitive to picrotoxin (*Figure 2F*). In some other neurons, an inward excitatory postsynaptic current sensitive to CNQX was recorded when the neurons were held at –70 mV (*Figure 2G*). Quantification of the presence of these light-evoked postsynaptic currents revealed the presence of both excitatory and inhibitory currents in 34.9% of neurons recorded, only inhibitory currents in 52.4% of neurons, and only excitatory currents in 12.7% (*Figure 2H*). These currents could be completely blocked by tetrodotoxin (TTX) but restored with the addition of 4-aminopyridine, indicating a mono-synaptic connection (an example shown in *Figure 2I–K*). Collectively, our findings demonstrate that heat-responsive POA neurons project directly to pPVT via both excitatory and inhibitory innervations.

## Activity changes of POA recipient pPVT neurons throughout chronic heat exposure

We speculated that repeated activation of POA recipient pPVT neurons underlie the emotional valence changes observed following chronic heat exposure. To gain support for this hypothesis, we first monitored the activity changes of these neurons during the course of chronic heat exposure by in vivo fiber photometry of calcium signals (*Figure 3A and B*). We injected AAV1-hSyn-Cre-EGFP into POA and Cre-dependent AAV9-hSyn-Flex-jGCaMP8F into pPVT and implanted optical fibers into pPVT neurons after enough expression of GCaMP. The fiber photometry recordings revealed fluctuations in signals composed of spontaneous events (details in Materials and methods) superimposed on the baseline. The detected spontaneous calcium events presumably represent brief trains or bursts of neuronal firing (*Ali and Kwan, 2020*). We first confirmed that the signals were stable for extended period (at least 21 days) in a group of mice not undergoing any treatment (*Figure 3—figure supplement 1*). As illustrated in *Figure 3C*, prior to subjecting the mice to chronic heat exposure, we recorded the neuronal activities of POA recipient pPVT neurons for each mouse as the pre-heat control. We then recorded the calcium events 1 day after acute heat exposure on day 1 and after chronic heat exposure on day 21. Interestingly, when compared with the pre-heat condition of day 1, both the frequency and amplitude during heat exposure on day 1 and day 21 were significantly increased (*Figure 3D and F*). However, such increases in calcium events disappeared on the day following heat exposure on day 1 indicating its transient nature (*Figure 3E*). In contrast, after 21 days of heat exposure, such increases in the frequency of calcium events persisted (*Figure 3G*). It is worth noting that there were no significant differences in both the frequency and amplitude of calcium events during heat exposure on day 1 and day 21 (*Figure 3H*).

Based on the increased neuronal activities of POA recipient pPVT neurons after chronic heat exposure, we investigated the impact of activating the pathway from the POA to pPVT on emotional valence and arousal levels of mice (*Figure 3—figure supplement 2A*). Utilizing the real-time place preference behavioral paradigm, we noticed that when the blue light was turned on, mice that had already gone through the habituation process quickly entered the chamber with the light turned off (*Figure 3—figure supplement 2B and C*). This finding suggests that the activation of the POA to pPVT circuit induces an aversive emotional valence. Notably, optogenetic activation did not affect

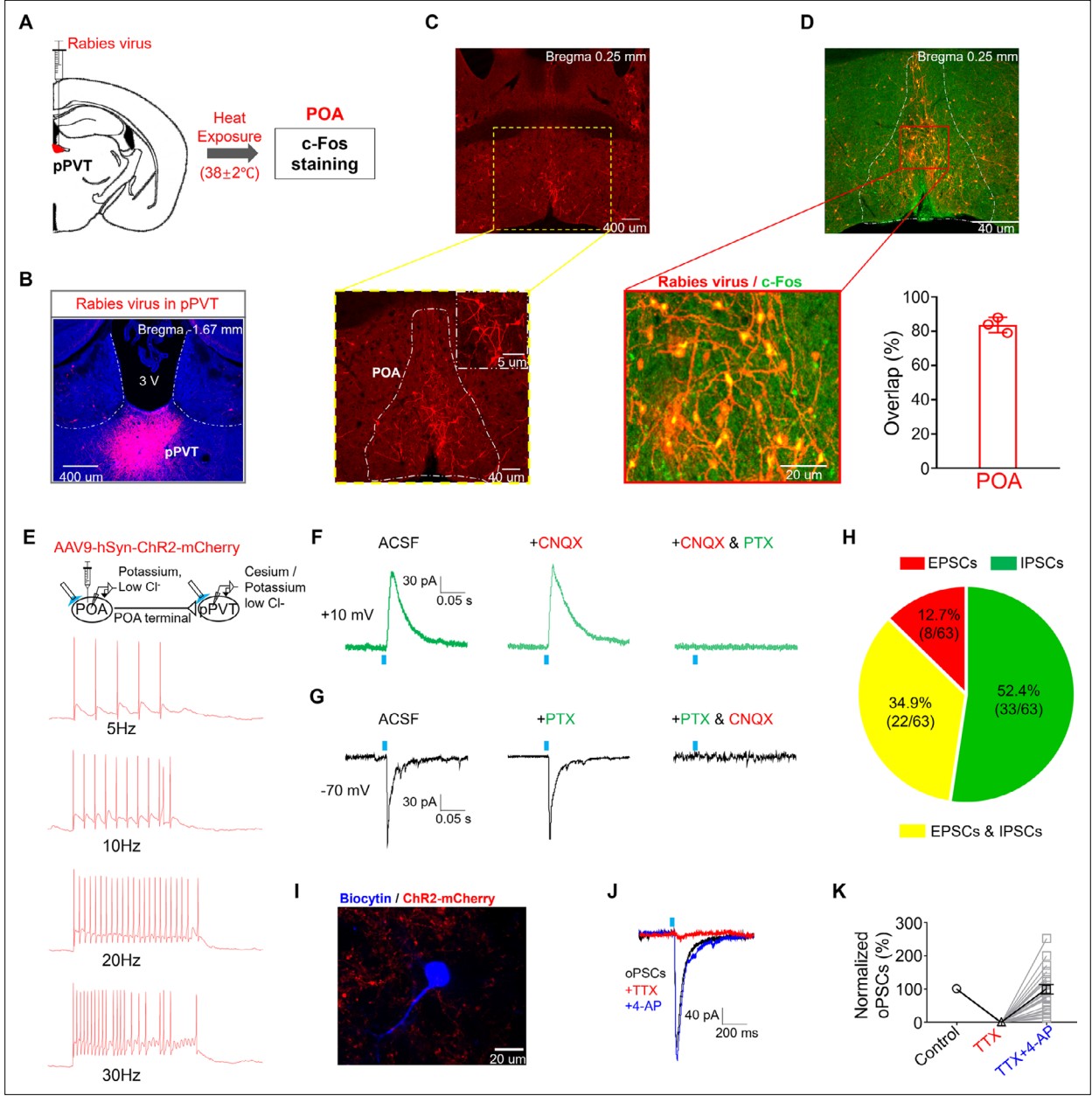

**Figure 2.** Involvement of the hypothalamic preoptic area (POA) to posterior paraventricular thalamus (pPVT) projections. (**A**) The strategy of virus injection followed by heat exposure-induced c-Fos staining (n=3 mice). (**B**) The representative microphotograph showed the expression of rabies virus in the pPVT regions. (**C**) Rabies virus retro-labeled POA neurons. Upper: magnification: ×4, scale bar: 400 μm. Lower: magnification: ×10, scale bar: 40 μm. The top right corner of the lower picture: magnification: ×60, scale bar: 5 μm. (**D**) The representative microphotograph and quantification showed that most of pPVT rabies virus retro-labeled POA neurons co-stained with heat exposure-induced c-Fos. Upper: magnification: ×10, scale bar: 40 μm. Lower and left: magnification: ×20, scale bar: 20 μm. (**E**) The strategy of virus injection (n=5 mice) and patch-clamp recording was performed on POA expressing ChR2-mCherry neurons and pPVT neurons using potassium and cesium, low chloride internal solutions, respectively. Representative traces showed that POA expressing ChR2-mCherry neurons exhibited robust firing in response to optical stimulation at different frequencies (n=15 neurons from 5 mice). (**F, G**) Representative traces showed that blue light stimulation evoked either excitatory postsynaptic current (EPSC) which could be blocked by cyanquixaline (CNQX, 10 μM) or inhibitory postsynaptic current (IPSC) which could be blocked by picrotoxin (PTX, 100 nM). (**H**) Pie chart showed the projection types recorded on pPVT neurons (n=63 neurons from 10 mice). (**I**) The representative recorded pPVT neuron was visualized by biocytin staining and was found being surrounded by POA expressing ChR2-mCherry terminals. Magnification: ×60, scale bar: 20 μm. (**J, K**) The representative trace showed that the application of tetrodotoxin (TTX) eliminated the oPSC held at –70 mV while the addition of 4-aminopyridine (4-AP, 1 mM) recovered it and the quantification (n=24 neurons from 10 mice).

The online version of this article includes the following figure supplement(s) for figure 2:

*Figure 2 continued on next page*

*Figure 2 continued*

**Figure supplement 1.** Posterior paraventricular thalamus (pPVT) was strongly activated after single-time heat exposure.

**Figure supplement 2.** Preoptic area (POA) recipient posterior paraventricular thalamus (pPVT) neurons were activated after heat exposure.

the movement activity of the mice during the experiment (*Figure 3—figure supplement 2D*). And we did not observe any change in core body temperature when stimulating the POA-pPVT circuit. On the other hand, as pupil size is widely recognized utilized for reflecting arousal levels in rodent models (*Privitera et al., 2020*), we further studied the effect of optogenetic activation of POA excitatory terminals in pPVT on pupil size of the mice under head fixation. The photos (*Figure 3—figure supplement 2E*) and video (*Figure 3—video 1*) clearly demonstrated the enlargement of both the pupil (*Figure 3—figure supplement 2F and G*) and eye sizes (*Figure 3—figure supplement 2H and I*) of the mice during light stimulation. Collectively, our results demonstrate that POA recipient pPVT neurons exhibited heightened activities after chronic heat exposure. Moreover, the activation of the POA to pPVT circuit is associated with negative emotional states and higher levels of arousal.

## POA recipient pPVT neurons are sufficient and necessary for chronic heat exposure-induced negative emotional valence and hyperarousal states in mice

To confirm the direct relationship between repeated activation of POA recipient pPVT neurons and the negative emotional and hyperarousal states in mice, we conducted chronic activation of POA excitatory terminals within pPVT by injecting *Camk2a*-promotor encoded ChR2 into the POA and implanting an optical fiber in pPVT (*Figure 4A*). We applied a chronic activation protocol, described by *Sidor and McClung, 2014*, which involved a cycle of 2 min ON and 2 min OFF, for a total of 20 min per day, over a period of up to 21 days. Following chronic optogenetic activation, we observed several behavioral changes in mice that were similar to those seen in mice exposed to chronic heat. Notably, in the elevated plus maze test, mice subjected to chronic optogenetic activation spent less time in the open arms (*Figure 4B and C*), representing increased stress levels. In the three-chamber test, these mice displayed reduced interest in exploring the unfamiliar male mouse compared to an inanimate object (*Figure 4D and E*), suggesting decreased sociability. During the female encounter test, these mice engaged less with the unfamiliar female mouse compared with the unfamiliar male mouse (*Figure 4F and G*), indicating decreased innate motivation. In the resident-intruder aggression test, the chronic heat-exposed mice exhibited both the decreased latencies in initiating the first attack and increased attack durations (*Figure 4H and I*), suggesting elevated aggression levels. Additionally, compared to the control group, mice after chronic optogenetic activation exhibited more pronounced body fluctuations when captured by a high-resolution camera in response to a 105 dB sound stimulus, indicating the heightened arousal levels (*Figure 4J–M*). Our results directly demonstrated that chronic optogenetic activation of POA excitatory terminals within pPVT circuit could induce negative emotional and hyperarousal states in mice.

To establish the necessity of POA recipient pPVT neurons for chronic heat exposure-induced emotional changes, we optogenetically inhibited the POA recipient pPVT neurons during chronic heat exposure by employing a cycle stimulation strategy of 3 mins ON followed by 3 mins OFF. Only mice with sufficient virus infection and accurate fiber implantation were included in the data analysis (*Figure 4O*). The optogenetic inhibition prevented chronic heat-induced behavioral changes, including anxiety levels (*Figure 4P and Q*). sociability (*Figure 4R*), innate motivation (*Figure 4S*), aggression levels (*Figure 4T and U*), and arousal state (*Figure 4V and W*). These findings further supported the essential role of POA recipient pPVT neurons in mediating chronic heat exposure-induced negative emotional and hyperarousal states.

## Chronically activated POA recipient pPVT neurons exhibit exaggerated response to stressful situations

To gain further insight into the roles of POA recipient pPVT neurons in stress-related behaviors, we elucidated the relationship of their calcium activities (*Figure 5A*) with specific behaviors indicative of emotional transitions and hyperarousal states. In the elevated plus maze test in which the chronic heat-exposed mice spent less time in the open arms (*Figure 1B and C*), there were also reduced

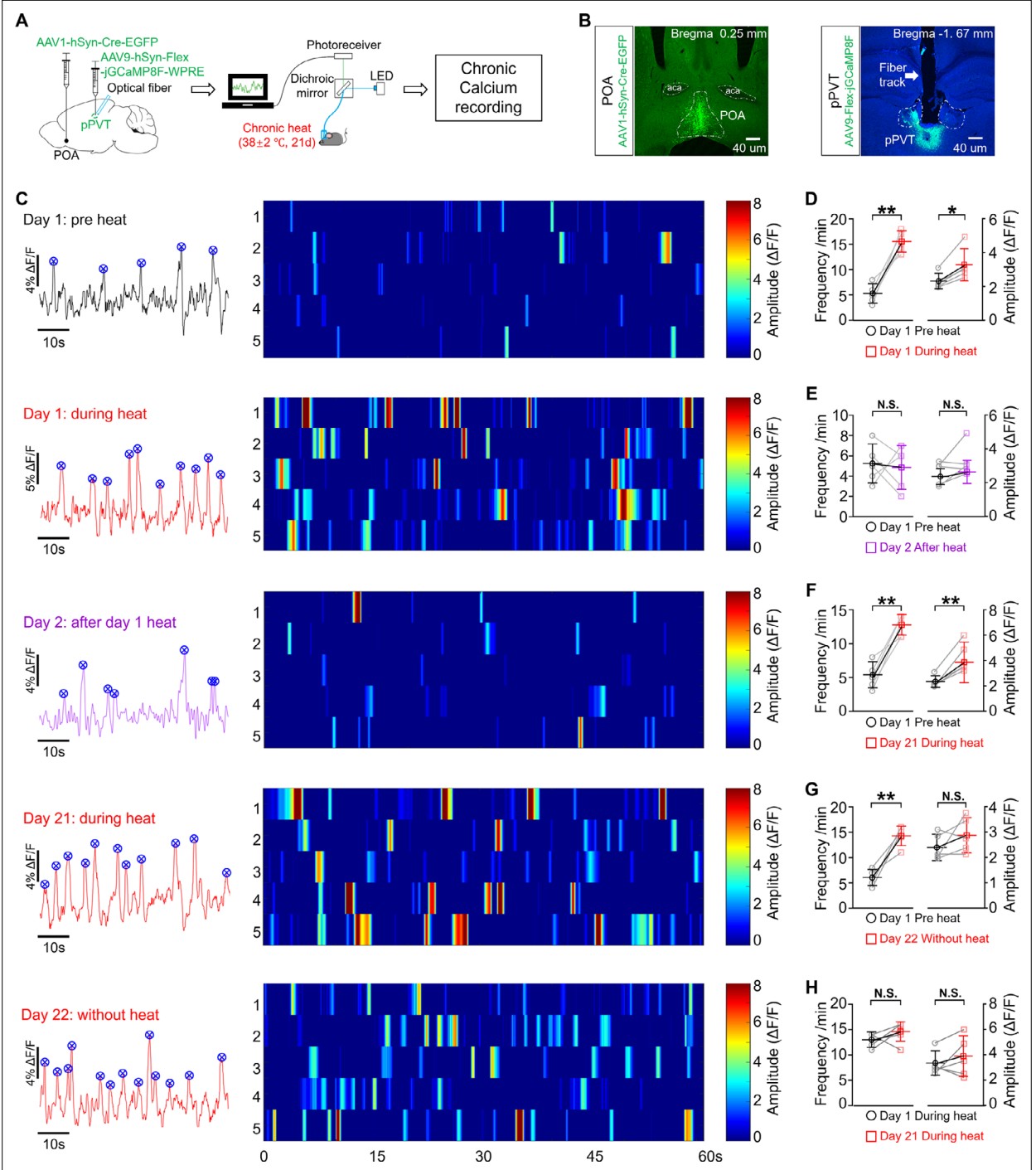

**Figure 3.** Activity changes of preoptic area (POA) recipient posterior paraventricular thalamus (pPVT) neurons throughout chronic heat exposure. (**A**) Experimental schematics. Mice (n=5) were stereotaxically injected with Cre-dependent GCaMP into the pPVT, followed by the implantation of optical fiber and chronic calcium recording. (**B**) Representative microphotographs showed the virus expression in the POA and pPVT regions, scale bar: all 40 μm. (**C**) From the top to the bottom: the representative calcium events (on the left panel) and the calcium events from different mice (on the right panel) on different days. Each vertical stripe in the heatmap represents one calcium event for each mouse. Statistical analysis of the frequency and amplitude of POA recipient pPVT neurons' calcium events compared for (**D**) pre and during heat exposure on day 1 (paired, parametric, two-tailed t-test; frequency (t=6.278, df = 4, p=0.0033); amplitude (t=4.344, df = 4, *p=0.0122)); (**E**) pre heat on day 1 and after heat on day 2 (paired, parametric, two-tailed t-test; frequency (t=0.2787, df = 4, p=0.7943); amplitude (t=1.726, df = 4, p=0.1595)); (**F**) pre heat on day 1 and during heat exposure on day 21 (paired, parametric, two-tailed t-test; frequency (t=6.124, df = 4, **p=0.0036); amplitude (t=4.704, df = 4, **p=0.0093)); (**G**) pre heat on day 1 and after chronic heat on day 22 (paired, parametric, two-tailed t-test; frequency (t=6.216, df = 4, **p=0.0034); amplitude (t=1.36, df = 4, p=0.2454)); (**H**) during

*Figure 3 continued on next page*

*Figure 3 continued*

heat exposure on day 1 and day 21 (paired, parametric, two-tailed t-test; frequency (t=1.242, df = 4, p=0.2821); amplitude (t=0.9424, df = 4, p=0.3993)). *p<0.05, **p<0.01, NS: not significant.

The online version of this article includes the following video and figure supplement(s) for figure 3:

**Figure supplement 1.** The calcium activities of the preoptic area (POA) recipient posterior paraventricular thalamus (pPVT) neurons were stable within our experimental period.

**Figure supplement 2.** Optogenetic activation of preoptic area (POA) excitatory neuronal terminals within posterior paraventricular thalamus (pPVT) produced aversive emotional valence and increased pupil size in mice.

**Figure 3—video 1.** Representative video showed the enlargement of both the pupil and eye sizes of the head-fixed mouse during blue light stimulation of preoptic area (POA) excitatory terminals within posterior paraventricular thalamus (pPVT).

https://elifesciences.org/articles/101302/figures#fig3video1

number of entries into the open arms (*Figure 5B*). More detailed analysis revealed that, compared with untreated mice, these animals more frequently paused at the center of the maze and then returned to the closed arms (*Figure 5C*). Furthermore, mice after chronic heat exposure exhibited a higher chance of running, rather than walking, back to the closed arms (*Figure 5D*) and their running speed significantly increased compared to their pre-heat condition (*Figure 5E*), presumably reflecting a state of increased anxiety. Interestingly, POA recipient pPVT neurons consistently exhibited a peak in activity when they were about to run back to the closed arm, while chronic heat-exposed mice exhibited a significantly larger peak (*Figure 5F–H*). Notably, no comparable increases of activities in the POA recipient pPVT neurons were observed when chronic heat-exposed mice paused at the center and finally walked out to the open arms (*Figure 5—figure supplement 1A–C*) or walked to the closed arms (*Figure 5—figure supplement 1D–F*). Similar phenomena were found with respect to the transitions from the open arm to the closed arm, not only in terms of number of transitions (*Figure 5I and J*), walking and running behavior (*Figure 5K*), and speed of running (*Figure 5L*), but also a larger calcium peak response aligned with the running behavior (*Figure 5M–O*).

In the chronic heat-treated mice, we also observed a context-dependent change in behavior reminiscent of the hyperarousal states when we introduced them back into the heat chamber. The mice exhibited apparent locomotion hyperactivity (*Figure 5P*), as evidenced by a notable increase in motion speeds exceeding 400 mm/s (*Figure 5Q*) and a higher frequency of fast running (*Figure 5S*). A clear time correlation was found between the heightened calcium activities in POA recipient pPVT neurons and the main component of locomotion hyperactivity, namely fast running under this condition (*Figure 5T–V*). In contrast, when being placed into a different chamber not associated with heat exposure, no obvious changes were observed among these chronic heat-exposed mice in their total distance traveled (*Figure 5—figure supplement 1G*), motion speed (*Figure 5—figure supplement 1H*), number of running exceeding 400 mm/s (*Figure 5—figure supplement 1I*), the frequency of fast running (*Figure 5—figure supplement 1J*), or the neuronal activities (*Figure 5—figure supplement 1K–M*). Collectively, our data suggest the heightened or exaggerated activities of POA recipient pPVT neurons after chronic heat exposure could serve as a driving force for behaviors manifesting negative emotional and hyperarousal states in response to stressful conditions.

## Increased pre- and postsynaptic excitability of pPVT neurons but saturated circuitry neuroplasticity capacity following chronic heat exposure

To explore the potential mechanisms underlying the heightened activities of POA recipient pPVT neurons after chronic heat exposure, we conducted in vitro slice recordings on pPVT neurons obtained from mice exposed to room temperature, acute heat, and chronic heat, respectively (*Figure 6A and B*). Both acute and chronic heat exposure significantly increased the amplitude of miniature inhibitory postsynaptic currents (mIPSCs) in pPVT neurons but there was no difference in mIPSCs frequency between the acute and chronic heat groups (*Figure 6C and D*). On the other hand, we observed a significant increase in the frequency of miniature excitatory postsynaptic currents (mEPSCs) specifically in pPVT neurons after chronic but not acute heat exposure (*Figure 6E and F*). These results suggest that while heat exposure could modulate both inhibitory and excitatory synaptic inputs onto pPVT neurons, there was a differential increase in presynaptic excitability of the excitatory pathway

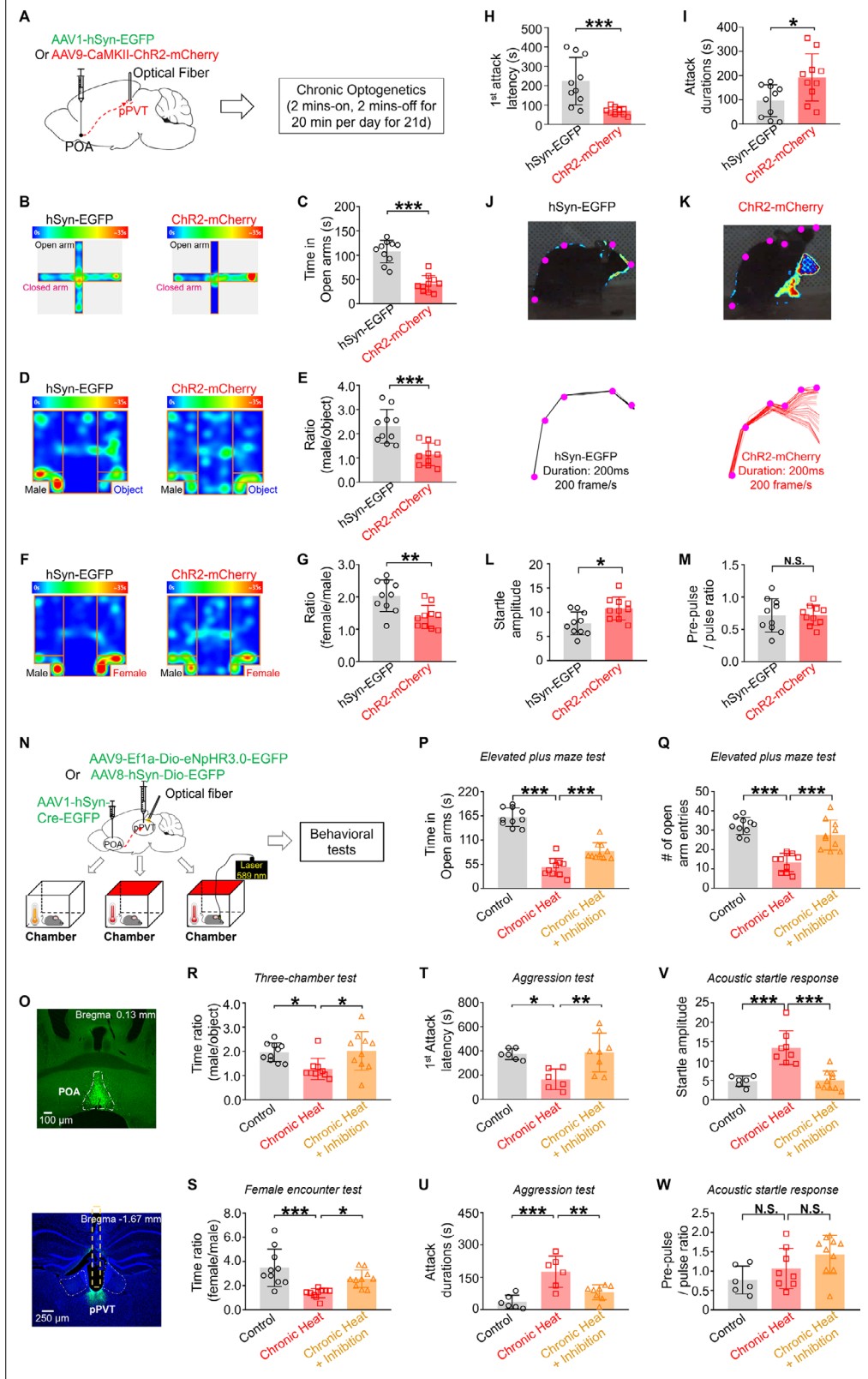

**Figure 4.** Preoptic area (POA) recipient posterior paraventricular thalamus (pPVT) neurons are sufficient and necessary for chronic heat exposure-induced negative emotional valence and hyperarousal states in mice. (**A**) Experimental schematics. Mice (n=10 in each group) were stereotaxically injected with either AAV1-hSyn-EGFP or AAV9-*Camk2a*-ChR2-mCherry into the POA, followed by the implantation of optical fiber into pPVT, chronic

*Figure 4 continued on next page*

*Figure 4 continued*

optogenetic activation, and behavioral tests. (**B, C**) The heatmap of representative tracking trace examples in elevated plus maze (EPM) test and the time spent in the open arms (Mann-Whitney unpaired two-tailed U test; U=2, ***p<0.001). (**D, E**) The heatmap of representative tracking trace examples in three-chamber test (TCT) and the interaction time with an unfamiliar male mouse relative to the inanimate object (Mann-Whitney unpaired two-tailed U test, U=8; ***p=0.0007). (**F, G**) The heatmap of representative tracking trace examples in female encounter test (FET) and the time surrounding the unfamiliar female mouse compared to an unfamiliar male mouse (Mann-Whitney unpaired two-tailed U test; U=13, **p=0.0039). (**H, I**) The first-time attack latency (Mann-Whitney unpaired two-tailed U test; U=8, p=0.007) and the attack durations (Mann-Whitney unpaired two-tailed U test; U=19, *p=0.0185) in the aggression test. (**J, K**) Visualized acoustic startle response (ASR) examples (upper panel) and corresponding labeled body parts' skeletons (lower panel) from hSyn-EGFP group and ChR2-mCherry group. (**L, M**) The startle amplitude (Mann-Whitney unpaired two-tailed U test; U=17, *p=0.0115) and the pre-pulse/pulse ratio (Mann-Whitney unpaired two-tailed U test; U=48, p=0.9118) in ASR test. (**N**) Experimental schematics. Mice (n≥6 in each group) were stereotaxically injected with AAV1-hSyn-Cre-EGFP in the POA and either AAV9-Ef1a-Dio-eNpHR3.0-EGFP or AAV8-hSyn-Dio-EGFP into the pPVT, followed by the implantation of optical fiber and behavioral tests. (**O**) Representative microphotographs showed the expression of AAV1-hSyn-Cre-EGFP and AAV9-hSyn-Dio-eNpHR3.0 within the POA and pPVT, respectively. Magnification: ×4, scale bar: 100 μm (upper panel) and 250 μm (lower panel). (**P, Q**) The time (one-way repeated measures ANOVA with Tukey post hoc test; $F_{(2, 27)}$=83.03, ***p<0.001; control vs. chronic heat, ***p<0.001; chronic heat vs. chronic heat+inhibition, ***p<0.001) and the entry numbers (one-way repeated measures ANOVA with Tukey post hoc test; $F_{(2, 20.92)}$=27.96, ***p<0.001; control vs. chronic heat, ***p<0.001; chronic heat vs. chronic heat+inhibition, ***p<0.001) into the open arms of EPM. (**R**) The time spent with an unfamiliar male mouse compared to an inanimate object in TCT (one-way repeated measures ANOVA with Tukey post hoc test; $F_{(2, 27)}$=5.381, *p=0.0108; control vs. chronic heat, *p=0.0292; chronic heat vs. chronic heat+inhibition, *p=0.0175). (**S**) The ratio of time with an unfamiliar female mouse compared to an unfamiliar male mouse in FET (one-way repeated measures ANOVA with Tukey post hoc test; $F_{(2, 27)}$=10.94, ***p=0.0003; control vs. chronic heat, ***p=0.0002; chronic heat vs. chronic heat+inhibition, *p=0.0335). (**T, U**) The first-time attack latency (one-way repeated measures ANOVA with Tukey post hoc test; $F_{(2, 17)}$=7.426, *p=0.0048; control vs. chronic heat, *p=0.0157; chronic heat vs. chronic heat+inhibition, *p=0.0063) and the attack durations (one-way repeated measures ANOVA with Tukey post hoc test; $F_{(2, 17)}$=13.38, ***p=0.0003; control vs. chronic heat, ***p=0.0003; chronic heat vs. chronic heat+inhibition, **p=0.0051) in the aggression test. (**V, W**) The startle amplitude (one-way repeated measures ANOVA with Tukey post hoc test; $F_{(2, 21)}$=21.03, ****p<0.0001; control vs. chronic heat, ***p<0.001; chronic heat vs. chronic heat+inhibition, ***p<0.001) and the pre-pulse/pulse ratio (one-way repeated measures ANOVA with Tukey post hoc test; $F_{(2, 21)}$=3.802, *p=0.039; control vs. chronic heat, p=0.2616; chronic heat vs. chronic heat+inhibition, p=0.2246) in ASR test. *p<0.05, **p<0.01, ***p<0.001, NS: not significant.

after chronic heat exposure but an increase in postsynaptic response to inhibitory input. Furthermore, when we examined the excitability of pPVT neurons by injecting a sequence of inward currents up to 100 pA to pPVT neurons, pPVT neurons from chronic heat-exposed mice exhibited significant increases in the number of action potentials in response to 30 to 70 pA injections (*Figure 6G and H*). Further analysis unveiled that these neurons exhibited a reduced rheobase for action potential generation (*Figure 6I*). However, no apparent alterations were observed in the other parameters (*Figure 6—figure supplement 1*).

The increase in presynaptic excitability of the POA to pPVT excitatory pathway suggested plastic changes induced by the chronic heat treatment. While it would be difficult to follow the excitability of this pathway in vivo during the chronic heat treatment, we sought to examine the synaptic plasticity capacity of this pathway before and after chronic heat treatment to shed light on its involvement. We injected *Camk2a*-promotor encoded ChR2 AAV into the POA of mice and divided them into two groups: a control group exposed to room temperature and a chronic heat group. After 21 days of heat exposure, sagittal slices that largely preserved the POA to pPVT pathway were prepared from both the control and chronic heat groups to induce long-term potentiation (LTP) (*Figure 6J*). Local optogenetic stimulation vertically (470 nm) through the objective was applied to elicit light-evoked EPSCs in POA recipient pPVT neurons and baseline of 10 min was recorded. High-frequency stimulation (HFS) is an effective induction protocol for eliciting LTP at excitatory synapses (*Zhu et al., 2016*; *Grover et al., 2009*; *Nugent et al., 2008*). Therefore, by delivering blue light at various frequencies (10, 30, 50, 70, 90 Hz) to activate the POA excitatory terminals within pPVT, we developed an optogenetic HFS protocol (HFS_opto: light pulses at 30 Hz, repeated three times at 20 s interval) effective in eliciting potentiated EPSCs on POA recipient pPVT neurons after baseline recording. HFS_opto protocol

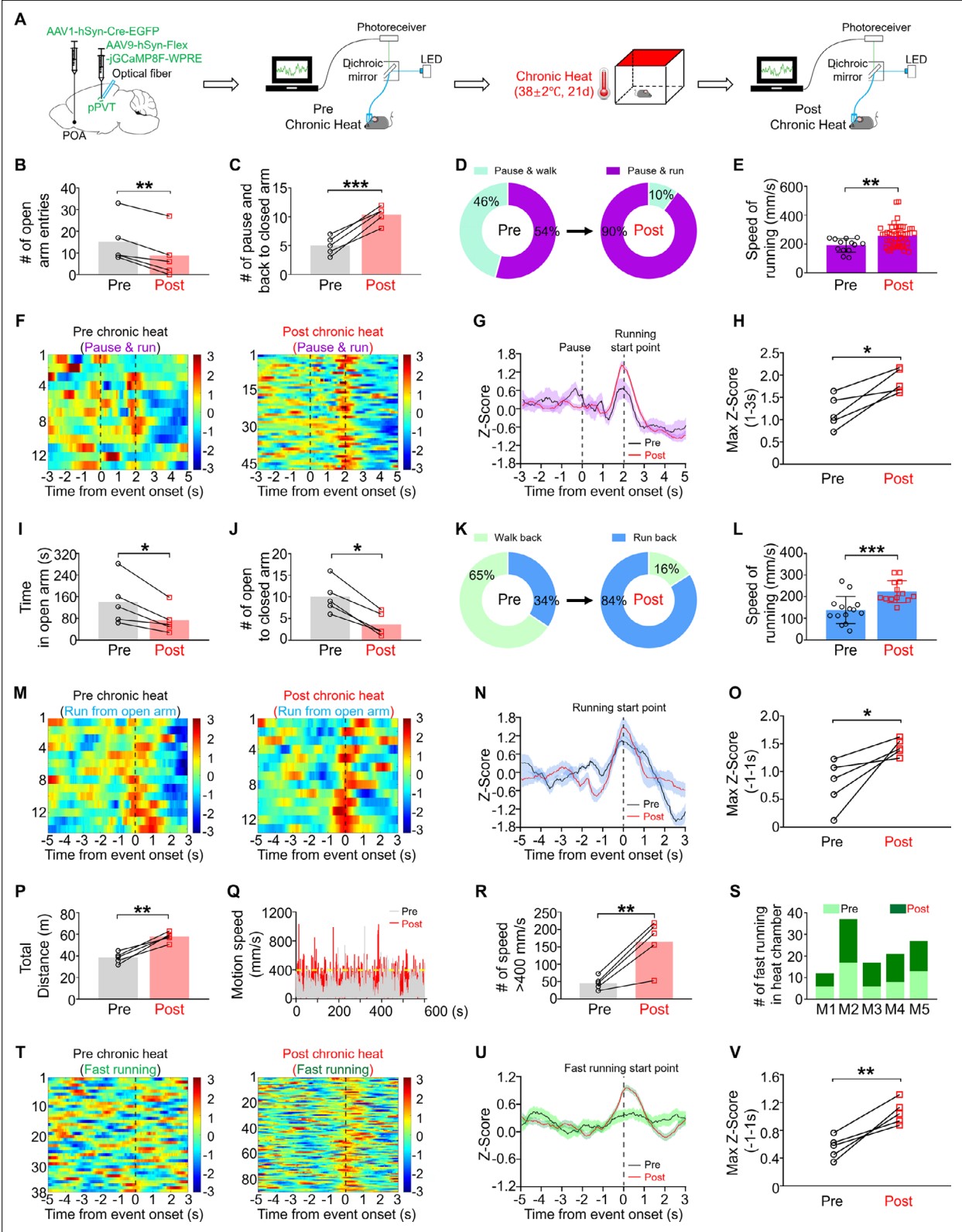

**Figure 5.** Chronically activated preoptic area (POA) recipient posterior paraventricular thalamus (pPVT) neurons exhibited exaggerated response to stressful situations. (**A**) Experimental schematics. Mice (n=5) were stereotaxically injected with AAV1-hSyn-Cre-EGFP in the POA and AAV9-hSyn-Flex-jGCaMP8F-WPRE in the pPVT, followed by the implantation of optical fiber and calcium recording. (**B**) The number of entries into the open arms of elevated plus maze (EPM) (paired, parametric, two-tailed t-test; t=6.901, df = 4, **p=0.0023). (**C**) The number of instances where the mice paused at the

*Figure 5 continued on next page*

*Figure 5 continued*

center area followed by back to closed arms of EPM (paired, parametric, two-tailed t-test; t=9, df = 4, ***p=0.0008). (**D**) Pie chart showed the changes of the percentage of pause and then running back to closed arms. (**E**) The changes of running speed in EPM (Mann-Whitney unpaired two-tailed U test; U=139, **p=0.0028). (**F–H**) Heatmaps showed the calcium activities of POA recipient pPVT neurons when mice performed the pause-and-run-back-to-closed-arms behavior for pre (n=13 trials from 5 mice) and post conditions (n=47 trials from 5 mice), Z-score calcium signals, and statistical comparison (paired, parametric, two-tailed t-test; t=4.387, df = 4, *p=0.0118). (**I**) The time spent in open arms of EPM (paired, parametric, two-tailed t-test; t=3.649, df = 4, *p=0.0218). (**J**) The total number of the behavioral event: back from open to closed arms (paired, parametric, two-tailed t-test; t=6.901, df = 4, **p=0.0023). (**K**) Pie chart showed the changes of the number of running episodes from open to closed arms for pre and post conditions. (**L**) The changes of running speed in EPM (Mann-Whitney unpaired two-tailed U test; U=23, ***p=0.0005). (**M–O**) Heatmaps showed the calcium activities of POA recipient pPVT neurons when mice performed fast running from open to closed arms for pre (n=14 trials from 5 mice) and post conditions (n=13 trials from 5 mice), Z-score calcium signals, and statistical comparison (paired, parametric, two-tailed t-test; t=3.087, df = 4, *p=0.0367). (**P**) The total distances of mice traveled in the chamber previously subjected to chronic heat (paired, parametric, two-tailed t-test; t=6.876, df = 4, **p=0.0023). (**Q**) The changes of motion speed for pre and post conditions from one representative mouse. (**R**) The number of instances with motion speed exceeding 400 mm/s (paired, parametric, two-tailed t-test; t=5.100, df = 4, ***p=0.007). (**S**) The number of fast running episodes for pre and post conditions. (**T–V**) Heatmaps showed the calcium activities of POA recipient pPVT neurons when the mice performed fast running in the previous chronic heat-exposed chamber (n=90 trials from 5 mice) compared to pre-heat condition (n=38 trials from 5 mice), Z-score calcium signals, and statistical comparison (paired, parametric, two-tailed t-test; t=5.456, df = 4, **p=0.0055). *p<0.05, **p<0.01, ***p<0.001.

The online version of this article includes the following figure supplement(s) for figure 5:

**Figure supplement 1.** Preoptic area (POA) recipient posterior paraventricular thalamus (pPVT) neurons did not exhibit obvious changes in calcium activities when mice performed pause and walked to open arms, or walked to closed arms in the elevated plus maze (EPM), or displayed fast running in a heat-exposure-unrelated chamber.

could successfully induce LTP at POA-pPVT synapses. The amplitude of light-evoked EPSCs in the control group following HFS$_{opto}$, at the onset, showed an obvious increase and gradually became stable potentiation, lasting at least 30 min (150%±18 of baseline, 14 neurons from 5 mice) (*Figure 6K and L*). However, when we applied the same protocol to the POA recipient pPVT neurons of mice after chronic heat exposure, typically no LTP could be induced (*Figure 6K–M*), suggesting that the pathway in these mice was already saturated during chronic heat exposure.

Taken together, our findings suggest enhanced excitatory inputs to pPVT neurons and increased membrane excitability of these neurons may underlie the behavioral changes observed in mice following chronic heat exposure.

## Discussion

In this study, we aimed to uncover the neurobiological basis of how extended periods of high temperature such as heatwaves impact on emotional processing in the mammalian brain. We demonstrated that chronic heat exposure led to negative emotional valence and hyperarousal states in mice, but not depression-like behavior. We discovered that hypothalamic POA recipient pPVT neurons in the thalamus mediate such emotional state changes via increased baseline activities and excitability in response to stressful conditions. Presumably, this is attributed to that repeated activation of the POA to pPVT circuit modulates both pre- and postsynaptic processes leading to their long-term changes.

Different from utilizing heating panels (*Venkatachalam and Montell, 2007*) to investigate heat-induced thermoregulation, our study employed an air heating strategy. This choice was motivated by the aim of replicating the realistic conditions induced by heatwaves. To ensure the well-being of the animals and prevent unnecessary physiological harm, we maintained temperatures around 38°C, as temperatures exceeding 40°C can lead to adverse effects on the body (*Deuis et al., 2017*). It is noted that our study differs from many previous studies that applied less high temperature but continuously over several days and weeks, which could have beneficial effects related to cardiovascular fitness, energy metabolism, and even cognitive functions (*Murray et al., 2022*). Our findings revealed that the protocol of chronic heat exposure at 90 min per day for 21 days did not impact the thermoregulatory function in mice. In the various emotional state-related behavioral tests we conducted, including the open-field, three-chamber, and female encounter test, we did not observe locomotion hyperactivity in mice. However, a distinct ASR was noted following chronic heat exposure. This may initially appear contradictory, as hyperarousal states are typically associated with hyperlocomotion (*Zhang et al., 2020*). However, POA recipient pPVT neurons exhibited heightened responses to stressful and aversive situations, which suggested that chronic heat exposure might primarily induce a state of

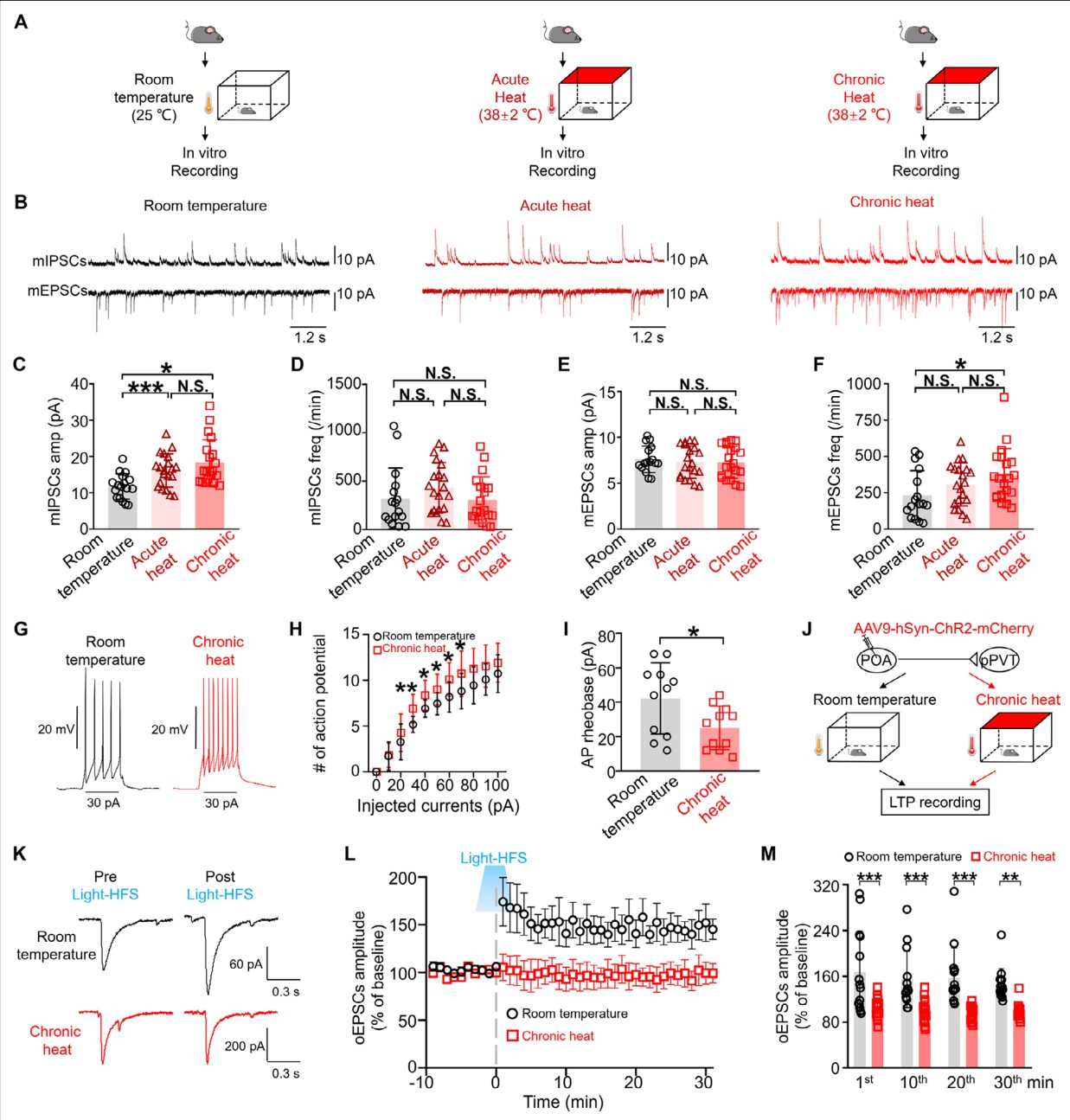

**Figure 6.** Increased pre- and postsynaptic excitability of posterior paraventricular thalamus (pPVT) neurons but saturated circuitry neuroplasticity capacity following chronic heat exposure. (**A**) Experimental schematics. In vitro brain slice recording was performed on mice from three groups (n=16 neurons from 3 mice in the room temperature group, n=20 neurons from 4 mice in the acute heat group, and n=20 neurons from 4 mice in the chronic heat group). (**B**) Representative traces of miniature postsynaptic currents from three groups. Duration: 12 s. Scale bar: 10 pA. (**C, D**) The changes of miniature inhibitory postsynaptic currents (mIPSCs') amplitude (one-way repeated measures ANOVA with Tukey post hoc test; $F_{(2, 54)}$=8.226, ***p=0.0008, room temperature vs. acute heat, ***p=0.0005; room temperature vs. chronic heat, *p=0.0268) and frequency (one-way repeated measures ANOVA with Tukey post hoc test; $F_{(2, 53)}$=1.346, p=0.269, room temperature vs. acute heat, p=0.4327; room temperature vs. chronic heat, p=0.9787) of pPVT neurons. (**E, F**) The changes of miniature excitatory postsynaptic currents (mEPSCs') amplitude (one-way repeated measures ANOVA with Tukey post hoc test; $F_{(2, 52.98)}$=0.1995, p=0.8198, room temperature vs. acute heat, p=0.8433; room temperature vs. chronic heat, p=0.8433) and frequency (one-way repeated measures ANOVA with Tukey post hoc test; $F_{(2, 50.1)}$=2.981, p=0.0598, room temperature vs. acute heat, p=0.2931; room temperature vs. chronic heat, *p=0.0473) of pPVT neurons. (**G**) The representative traces of action potential of pPVT neurons upon 100 pA current injection. (**H**) The changes of excitability of pPVT neurons when different currents were injected to the patched pPVT neurons (n=11 neurons from 3 mice in each groups; Mann-Whitney unpaired two-tailed U test; 30 pA: U=22.5, **p=0.0091; 40 pA: U=31, *p=0.048; 50 pA: U=29.5, *p=0.0412; 60 pA: U=25, *p=0.0167; 70 pA: U=29.5, *p=0.0396). (**I**) The changes of rheobase of action potential (Mann-Whitney unpaired two-tailed U test; U=28.5, *p=0.0345).

*Figure 6 continued on next page*

*Figure 6 continued*

(**J**) Experimental schematics. Mice (n=5 mice in each group) stereotaxically injected with AAV9-*Camk2a*-ChR2-mCherry into the preoptic area (POA) were then divided into the room temperature and chronic heat groups. Sagittal slices of mice were prepared for long-term potentiation (LTP) induction and recording. (**K**) The representative traces showed pPVT neurons from mice exposed to room temperature and chronic heat exhibited different amplitude of oEPSCs after blue light-mediated high-frequency stimulation. (**L**) The LTP induction and recording of POA to pPVT pathway from slices of mice exposed to room temperature and chronic heat conditions after blue light stimulation at 30 Hz (n=14 neurons from control group and 18 neurons from chronic heat group). (**M**) Statistical comparison of the amplitude of oEPSCs at different time points (two-way repeated measures ANOVA with Sidak post hoc test; interaction: $F_{(3, 120)}=0.3486$, $p=0.7902$; optical stimulation main effect: $F_{(3, 120)}=76.2$, \*\*\*$p<0.001$; time points effect: $F_{(3, 120)}=0.8215$, $p=0.4844$; 1st min: room temperature vs. chronic heat: \*\*\*$p<0.001$; 10th min: room temperature vs. chronic heat: $p=0.0001$; 20th min: room temperature vs. chronic heat: \*\*\*$p<0.001$; 30th min: room temperature vs. chronic heat: \*\*$p=0.0024$). \*$p<0.05$, \*\*$p<0.01$, \*\*\*$p<0.001$, NS: not significant.

The online version of this article includes the following figure supplement(s) for figure 6:

**Figure supplement 1.** The effect of chronic heat exposure on intrinsic properties of posterior paraventricular thalamus (pPVT) neurons.

increased susceptibility toward anxiety and hyperarousal rather than persistent hyperarousal states in mice.

There are a number of studies revealing the connections from the POA to pPVT. Some previous reports have already shown that both excitatory and inhibitory neurons within POA contribute to thermoregulatory function (*Morrison and Nakamura, 2011*). Augustine et al. on the other hand demonstrated that POA neurons expressing nNOS send both excitatory and inhibitory projections to PVT, with an emphasis on their roles in thirst regulation (*Augustine et al., 2018*). In contrast, *Allen et al., 2017*, previously described those thirst-related POA neurons sending projections to PVT-mediated reinforcing behavior, highlighting their independence from POA heat-responsive neurons. The findings of the present study enriched our understanding that heat-responsive neurons in the POA could send both excitatory and inhibitory projections to pPVT, and uncovered the role of excitatory projections from POA to pPVT in chronic heat exposure-induced emotional changes. While the role of the excitatory inputs in mediating the negative emotional valence and hyperarousal state is unveiled in this study, the role of the inhibitory inputs awaits further investigation.

Notably, we observed that POA recipient pPVT neurons exhibited higher baseline activities and displayed heightened response to stress or aversive stimuli following chronic heat exposure. Consistently, a recent study also found that long-term heat exposure rendered POA neurons to become tonically active (*Ambroziak et al., 2025*). Interestingly, prior studies have reported that PVT neurons exhibited stronger activation when mice were subjected to novel stimuli after repeated restraint stress (*Vertes et al., 2015*; *Bhatnagar and Dallman, 1998*; *Heydendael et al., 2011*). This also effectively supports our theory of heightened susceptibility to anxiety and our results provide a rationale for this widely observed phenomenon. Meanwhile, the specific neuronal signals correlated with actions indicative of heightened anxiety such as fast running to closed arms and fast running in chronic-heat exposed chamber were not induced by the motion per se. Our recording was conducted when mice were spontaneously behaving without any pre-set cues, excluding the influence of salient stimuli on the activity of POA recipient pPVT neurons (*Penzo and Gao, 2021*).

In addition to the enhanced excitatory connections from POA to pPVT after chronic heat exposure, we observed an increased amplitude of inhibitory postsynaptic response after both acute and chronic heat exposure. Increased inhibition could act to compensate for the initial over-excitation at critical time, to ensure that neurons briefly activated by stress can be effectively regulated and prevent the occurrence of excessive stress-related behaviors (*Perica and Luna, 2023*). pPVT neurons are anatomically innervated by strong inhibitory projections from different brain nuclei (*Penzo and Gao, 2021*). Acute stress-mediated disinhibition of pPVT neurons, as reported by *Beas et al., 2018*, can rapidly induce an anxiety-like state in mice. Other studies also showed that inhibition of activated neurons induced by stressful stimuli is effective to avoid abnormal affective behavioral performances (*Cruz et al., 2024*; *Ghasemi et al., 2022*). Our results showed that long-term optogenetic inhibition of POA recipient pPVT neurons during chronic heat exposure can effectively prevent changes of emotional and hyperarousal states in mice. Therefore, the increase in mIPSC amplitude observed under both acute and chronic heat conditions in this study suggests the potential presence of negative feedback regulation in response to the over-excitation. However, under chronic heat exposure, further persistent significant increase in excitatory synaptic transmission may surpass this effect, leading to a final net activation in the pPVT neurons. However, the potential mechanisms by which increased

excitatory synaptic transmission and enhanced postsynaptic inhibitory responses interact to achieve activation or inhibition of neurons themselves still require further study.

Current hypothesis regarding the mechanism of pPVT in mediating anxiety-like behavior primarily focused on the dopamine-mediated disinhibition of hypothalamic inhibitory terminals within pPVT during acute stress (*Beas et al., 2018*). Our study distinctly revealed that the enhanced excitatory inputs to pPVT and increased excitability following chronic heat exposure are responsible for the ensuing negative emotional and hyperarousal states. Interestingly, different from previous studies that reported a decrease in the amplitude of mIPSC in pPVT-expressing dopamine receptor 2 neurons 24 hr after acute restraint stress (*Beas et al., 2018*), our slice recording from pPVT neurons 24 hr after acute heat exposure exhibited an increase in mIPSCs amplitude. The reasons behind these differences could be attributed to the treatment applied and the specific neuronal types under investigation. Both the enhanced presynaptic excitability and increased intrinsic postsynaptic excitability following chronic heat exposure suggested the involvement of long-term neuroplasticity. The fact that LTP of the POA to pPVT pathway was no longer inducible after chronic heat exposure strongly suggested that this pathway was potentiated during the heat treatment, which accounted for the increased excitability of POA terminals to pPVT and their increased neuronal activities. Overall, both the enhanced excitatory inputs to pPVT and increased excitability may play an indispensable role in chronic heat exposure-induced emotional changes in mice.

There are certain limitations in the scope of the present study. First, our study was conducted solely on male mice. Further investigation is required to determine whether chronic heat exposure can elicit similar behavioral phenotypes in female mice. Also, the pPVT is known to express various peptide receptors (*Curtis et al., 2020*). However, the specific roles of these peptide receptors in chronic heat exposure-induced emotional changes remain unexplored. Some possible mechanisms may contribute to heightened activities of POA recipient pPVT neurons, such as the increased sensitization, the increased activity of HPA axis, the transition from chronic adaptation to pathological condition (*Herman, 2013*), and even the chronic effect of dopamine-induced disinhibition of inhibitory terminals within pPVT, all of which require further investigations. Moreover, POA and pPVT were both previously reported to be activated by stimuli other than heat (*Penzo and Gao, 2021*; *Zhang et al., 2021*). Additional studies are needed to explore the specificity of POA recipient pPVT neurons on chronic heat-induced emotional changes.

In conclusion, our findings provide compelling evidence that the heightened activities of POA recipient pPVT neurons are critically involved in chronic heat-exposure-mediated negative emotional valence and hyperarousal states, and render mice to enter anxiety-like state more readily, as summarized in *Figure 7*. These results shed new light on the mechanisms underlying heatwaves-induced emotional changes.

## Materials and methods

### Animals

Adult C57BL/6 male mice (25–45 g) were used in this study. Animals were bred and maintained by the Laboratory Animal Service Centre of The Chinese University of Hong Kong (CUHK). All experiments were performed following the CUHK guideline approved by the Animal Experimentations and Ethics Committee.

### Stereotaxic surgery and viruses

Mice were anesthetized with ketamine and xylazine and placed gently in a stereotaxic frame (Narashige, Tokyo). A Hamilton syringe (33-gauge) filled with AAV was placed into target brain areas according to the corresponding coordinates: POA (+0.62 mm A/P, 0 mm M/L, 5.83 mm D/V), pPVT (–1.47 mm A/P, 0 mm M/L, 3.13 mm D/V) from brain skull. 0.10–0.35 µl viruses were injected at 10 nl/min speed. After injection, the needle was left in the targeted brain area for an additional 10 min before retraction. The following viruses were used: AAV1-hSyn-Cre (Addgene, retrograde tracing), AAV1-hSyn-Cre-EGFP (Addgene, optogenetics, photometry recording), Rabies virus (BrainVTA, retrograde tracing), AAV9-hSyn-ChR2-EGFP (Addgene, functional connectivity), AAV9-hSyn-Flex-jGCaMP8F-WPRE (Addgene, photometry recording), AAV9-*Camk2a*-ChR2-mCherry (Addgene,

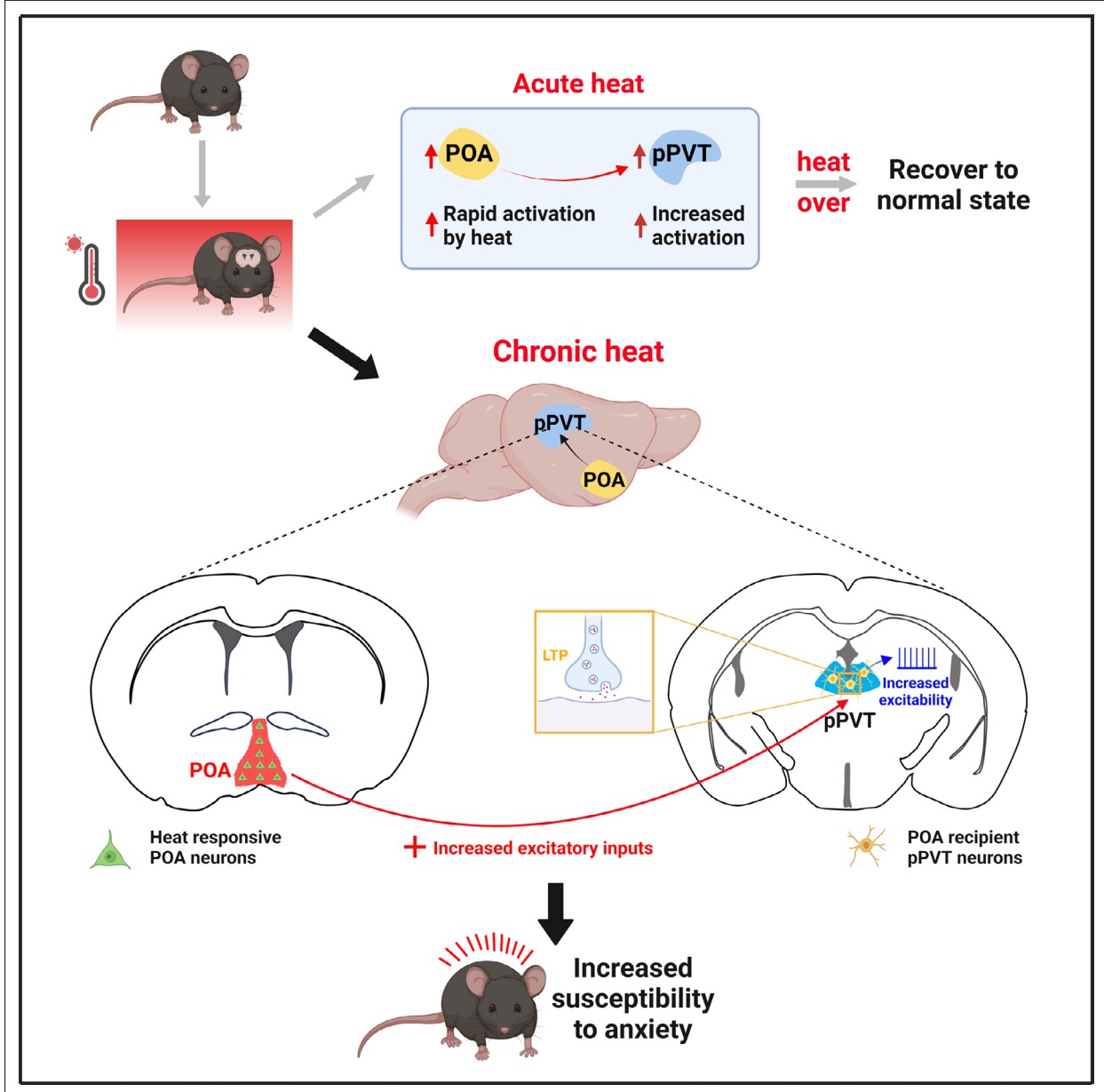

**Figure 7.** A working model of neural circuit mechanisms underlying preoptic area (POA) recipient posterior paraventricular thalamus (pPVT) neurons-mediated chronic heat exposure-induced negative emotional valence and hyperarousal states. Different from acute heat exposure, chronic heat exposure-induced enhancement in excitatory inputs to pPVT and saturated neuroplasticity contributed to the increased membrane excitability underlie the heightened activities of POA recipient pPVT neurons, rendering mice become more susceptible to stressful situations manifested as negative emotional valence and hyperarousal states.

optogenetic activation), AAV9-hSyn-Dio-eNpHR3.0-EGFP (Addgene, optogenetic inhibition), AAV9-hSyn-Dio-EGFP (Addgene, control experiments). All AAV titers were >5 × 10$^{12}$ particles per ml.

## Protocols for chronic heat exposure

The temperature (38±2°C) was adopted based on other reports (*Bridges et al., 2012*; *Cassuto, 1968*; *Cure, 1989*; *Jin et al., 2011*; *Rousset et al., 1984*). A home-made chamber was constructed for chronic heat exposure (80 cm [L]×40 cm [W]×60 cm [H]) which was composed of two built-in components (heating system and an environmental monitoring sensor). For the heating system, a heater with (1) adjustable buttons that could control the heating power and (2) a 'dual heating' setting that was able to stop heating when the temperature exceeded 40° in the chamber and restarted heating

when the temperature in the chamber decreased to lower than 36°C. After the air temperature of the chamber was gradually heated to the targeted temperature ranges and was monitored to be stable for at least 30 min, mice would be gently put into the chamber for 90 min of heat exposure per day for 21 days. During heat exposure, mice were freely accessible to drinking water and consuming food. The time point within each day to start heat exposure was random. For the environmental monitoring sensor, it was used to monitor the chamber temperature value and humidity level (Peng He Dian Zi, Shanghai, China). And the behavioral tests were conducted the following day after 21 days' heat exposure.

The method for measuring the body temperature of mice was based on a previous report (*Kawakami et al., 2018*). Briefly, prior to exposing them to heat, the mice were gently held by their tails to expose their lower abdomen. An infrared thermometer sensor was positioned beneath the lower abdomen and the temperature was measured. This measurement process was repeated three times for each mouse, and the stable values obtained were then averaged to determine the specific mouse's body temperature.

## Behavioral experiments

A variety of emotion-related behavioral tests were conducted on mice after acute and chronic heat exposure.

### Elevated plus maze test

The self-made elevated plus maze apparatus was made up of two open arms (25 cm [L]×5 cm [W]×0.5 cm [H]) and two closed arms (25 cm [L]×5 cm [W]×16 cm [H]) arranged in a 'plus' shape and elevated 50 cm above the floor. The mice were gently placed at the junction of the open and closed arms while facing toward one of the open arms to initiate the test. Mice were allowed for a 6 min exploration, and their behavior was videotaped and then analyzed and quantified by the ANY-Maze tracking software (Version 4.7, Stoelting CO).

### Three-chamber test and female encounter test

The chambers for both the three-chamber test and female encounter test were self-made by using an open-field arena (50 cm [L] × 40 cm [W] × 30 cm [H]) with two identical transparent chambers (40 cm [L] × 1 cm [W] × 20 cm [H]) with holes in the middle surface, the stranger male and female mouse were handled for 3 min and then habituated in a wire cage placed in the three-chamber apparatus for 5–10 min for 4 consecutive days. The tested mouse was habituated to the three-chamber apparatus for 10 min with all chambers unobstructed when the handling was over. The tested mouse was then guided to the center chamber when the habituation was over, and bidirectional exits were blocked. A wired cup (of size 0.5 cm [L] × 0.5 cm [W], which prevents large body contact and mating behavior but only induces innate sex-motivated exploratory movement around the cup) with a stranger mouse and an empty cup was introduced into the other two chambers, respectively, and then all chambers were opened for a sociability test. For the female encounter test, a wired cup with a strange male mouse and a wired cup with a strange female mouse was introduced into the left and right chambers, respectively, and all chambers were then opened for a motivation test. Their behavior was videotaped and then analyzed and quantified by the ANY-Maze tracking software (Version 4.7, Stoelting CO).

### Resident-intruder aggression test

Mice were fed in a single cage, in which a female mouse, separated by a wire mesh screen, was put in the corner of each cage for 7 days to facilitate each tested mouse to build their territory. After chronic heat exposure, the unfamiliar intruder mouse with a color label on the back was introduced to the cage of each tested mouse, and the video was recorded for 15 min. Each tested mouse's first attack latency and total attack durations were calculated and counted manually and double-blind by another lab mate.

### ASR test

The ASR test was conducted based on a recent study (*Pantoni et al., 2020*). In our study, a sound-proof box was purchased, in which a trumpet, a high-speed camera (200 frames per second, Plexon),

and a box made from acrylic plate (10 cm [L] × 5 cm [W] × 25 cm [H]) were prepared. Mice exposed to sound at 105 dB measured by a decibel meter exhibited obvious startle response while 75 dB evoked no response. 105 dB was thus set as the sound decibel of pulse-only stimulation with 0.2 s duration, while 75 dB was set as pre-pulse stimulation with 0.2 s duration. Before the ASR test, mice were placed into an acrylic box for 3 days of environmental habituation and then for the test. Pre-pulse inhibition (PPI) test started with at least 20 min baseline to make the tested mice quiet and immobile, and then the habituation phase, which consisted of 10 pulses, each 20 s apart, and finally presented with a PPI phase, in which 20 trials, 5 pulse-only trials and 15 pre-pulse and pulse trials, were delivered. In the pre-pulse and pulse trials, the pre-pulse was presented about 500 ms before the pulse stimulus. The mice's startle response was captured by a high-speed camera and then analyzed with the machine learning method: DeepLabCut. The brief details were as follows: the mice's body parts were manually labeled and labeled data were input into the google co-lab for the machine learning process until the learning error was lower than 0.5. The successful training data were exported as a .csv file. The startle amplitude of mice was calculated as the average value of all the labeled body parts' vibration amplitude within 200 ms after the sound was broadcast. Each trial was manually checked, and startle response affected by the mice's random moving would not be collected for further analysis. The startle response was visualized with custom MATLAB scripts.

The formula for calculating the PPI ratio was:

PPI ratio = amplitude of pre-pulse and pulse trial/amplitude of pulse-only trial.

## Open-field test

Mice were gently placed in the center of the open-field test chamber (60 cm [L]×60 cm [W]×40 cm [H]) and allowed for free exploration for 10 min. Mice's behaviors were analyzed and quantified by the ANY-Maze tracking software (Version 4.7, Stoelting CO).

## Forced swim test

The apparatus of the forced swim test was a cylindrical tank that was 30 cm high and 20 cm in diameter. And the water was 15 cm tall at room temperature (23–25°). Tested mice were gently put into water, and the video was recorded for 6 min. Mice's immobile time in the water was analyzed and quantified by the ANY-Maze tracking software (Version 4.7, Stoelting CO).

## Sucrose preference test

Mice from different groups were habituated for 2 days for two bottles (one is a sucrose solution bottle, the other is a water bottle, with left and right random placement) in their home cages. After habituation, the sucrose preference of mice from both groups was tested. The intake volume of water and 2% sucrose solution was measured within 2 hr. Sucrose preference was determined as the ratio of the intake volume of sucrose solution to total water consumption.

## Tail suspension test

The tested mice were gently grabbed with the tail and invertedly suspended in front of the recording camera by fixing their tail to the adhesive tape for 6 min of recording. Mice immobile time during the suspension was analyzed and quantified by the ANY-Maze tracking software (Version 4.7, Stoelting CO).

## Immunohistochemistry

Immunostaining for c-Fos was conducted on four distinct groups of mice: those exposed to acute heat, chronic heat, those injected with rabies virus into the pPVT and exposed to heat, and those with virus-labeled POA recipient pPVT neurons and exposed to heat.

## Retrograde tracing

The rabies virus system was adopted for retrograde tracing, as previously reported (*Watabe-Uchida et al., 2012*). On day 1, 50 nl of AAV1-hSyn-cre, 50 nl of AAV-FLEX-TVA-mCherry, and 50 nl of AAV-FLEX-RG were injected into pPVT. Subsequently, 80 nl of SAD△G virus was injected into pPVT on day 10. Mice were then allowed 7 days for full expression before being sacrificed.

## Tissue preparation

For mice from different groups, brain samples were prepared as follows: 40–60 min after exposure to heat, mice were anesthetized with i.p. injection of a ketamine-xylazine cocktail and transcardially perfused with 1× phosphate-buffered saline (PBS, Invitrogen) and then 4% paraformaldehyde (PFA, Sigma-Aldrich). The brain was then extracted and postfixed overnight in 4% PFA and finally dehydrated with 30% glucose solution for 48 hr. The well-prepared brain samples were embedded in OCT (Thermo Fisher) and sliced into 30 μm coronal sections by a cryostat (Leica) and stored in 1× PBS before immunohistochemistry.

## Immunofluorescent staining

Brain sections were blocked in 5% normal goat serum (Thermo Fisher) in PBS with 0.3% Triton X-100 (Sigma-Aldrich) for 40 min and then incubated with primary rabbit anti-c-Fos antibody (1:2000, Cell Signaling Technology) at 4°C overnight. After several thorough washing pieces in PBS, the sections were incubated with goat anti-rabbit IgG-Alexa Fluor 488 secondary antibody (1:1000, Invitrogen) in a blocking solution for 2 hr. After that, the sections were rinsed in PBS again and finally mounted onto glass slides. Microscopic images were taken under a confocal laser scanning microscope (C1, Nikon).

## Neuronal counting

As for the statistical analysis of c-Fos positive neurons, profile counting based on the rectangular square drawn in each picture was performed on ImageJ. For every counted brain area, c-Fos positive neurons on three consecutive brain slices were selected and measured. And the value averaged from profile counting on three consecutive brain slices was regarded as the c-Fos positive neuronal number of specific brain nuclei for one mouse.

## Optogenetic experiments

For activation: 100 nl of AAV9-*Camk2a*-ChR2-mCherry was injected into POA, optical fibers in pPVT. For inhibition: 100 nl of AAV1-hSyn-Cre-EGFP was injected into POA, 200 nl of AAV9-hSyn-Dio-eNpHR3.0-EGFP or AAV9-hSyn-Dio-EGFP in pPVT.

## Real-time place preference test

Mice were placed into a two-chamber box for 10 min of habituation to make sure that mice had no preference for either side of the chamber. Another 10 min of recording was started. The left chamber was set as the light ON compartment while the other was the light OFF chamber. Once mice entered light ON chamber, 473 nm light with 10 ms, 10 Hz (unless otherwise indicated, Newdoon Technology) was delivered via an optic cable (200 μm core, 0.37 NA, Doric Lens) while the light was turned off when mice entered light OFF chamber. Laser power was 5 mW measured at the tip of the fiber, which was implanted 0.1–0.2 mm above the targeted nucleus.

## Elevated plus maze test, three-chamber test, and female encounter test

To explore the effect of optogenetic activation of POA excitatory terminals within pPVT on the levels of anxiety, sociability, and motivation in mice, mice in these three behavioral tests were recorded with 3 min as the baseline, followed by another 3 min of delivery of 473 nm light stimulation via an optic cable (200 μm core, 0.37 NA, Doric Lens).

## Measurement of pupil size

To directly reflect the effect of POA excitatory terminals within pPVT on hyperarousal levels, mice were head-fixed and 473 nm light was delivered with 10 Hz, 10 ms at the mode of 10 s light off, 10 s light on, and 10 s light off to measure the changes of pupil size. And pupil size of mice during the whole process was synchronously captured through a high-speed camera at 30 frames per second. All pictures were finally integrated into video format through the FFmpeg program. The changes in pupil size among established videos were further analyzed by the DeepLabCut method (*Mathis et al., 2018*) for precise tracking and plotting. The time point for statistical analysis of normalized pupil size was at 5 s for pre, 15 s for light, and 25 s for post, respectively.

## Chronic optogenetic stimulation

Mice are subjected to 473 nm light stimulation with 10 ms, 10 Hz (unless otherwise indicated, Newdoon Technology) via an optic cable (200 μm core, 0.37 NA, Doric Lens) 2 min-on, 2 min-off for 20 min per day for up to 21 days (*Sidor and McClung, 2014*). Laser power does not exceed 5 mW measured at the tip of the fiber.

## Optogenetic inhibition

Mice exposed to chronic heat were synchronously illuminated with a 589 nm light laser (Newdoon Technology) to inhibit POA recipient pPVT neurons specifically. The stimulus was 20 mW with a cyclical mode consisting of 3 min of light continuous on and 3 min of light off, which was conducted for 21 days only during the heat exposure period (90 min per day) for each mouse. The emotion-related behaviors and hyperarousal states were measured as formerly mentioned and analyzed by ANY-Maze software.

## Fiber photometry

100 nl of AAV1-hSyn-Cre-EGFP in POA, 200 nl of AAV9-hSyn-Flex-jGCaMP8F-WPRE in pPVT, optical fibers in pPVT. Calcium signals were recorded for 1 min for each mouse before, during heat exposure on day 1, the second day after heat exposure on first day, and after chronic heat exposure on day 22. The frequency and amplitude of calcium events were collected for data analysis.

Before and after chronic heat exposure, the performance of each mouse in the elevated plus maze and the chronic heat-exposed chamber was in combination with calcium recording for 10 min, respectively. Targeting of event-related time points and calculation of speed were achieved through the SLEAP machine learning method (*Pereira et al., 2022*).

Fiber photometry data were collected with a TDT system at a sampling frequency of 1017 Hz. The LED power at the tip of the patch cord was less than 20 μW. The 405- and 465-signals were simultaneously recorded. The isosbestic 405 nm control signal was filtered using a polyfit regression to limit the influence of fluorescence decay during the session. And the fitted control signal was then subtracted from the 465-signal to remove artifacts from the intracellular calcium-dependent GCaMP fluorescence. The calculation for the dynamics of fluorescence in chronic calcium recording was performed using the formula $\triangle F/F=(F_{465}-F_{405})/F405$. As corroborated by previous studies (*Shao et al., 2022*; *Xia et al., 2021*), a calcium signal wave exceeding $\mu+3\sigma$ was regarded as a fluorescence transient. Here, $\mu$ and $\sigma$ respectively denote the average and the standard deviation of the baseline signal of each mouse. The baseline signal for each mouse was randomly selected, with a duration ranging from 5 to 10 s. As for the activity changes of POA recipient pPVT neurons during emotional transitions and hyperarousal states, the $\triangle F/F$ was then analyzed using a Z-score relative to the mean and standard deviation of the session. Data were visualized and analyzed with custom MATLAB scripts.

## Electrophysiological recordings

### Brain slice preparation and whole-cell recording

As for the confirmation of functional connectivity, coronal mice brain slices were prepared as follows. Mice quickly anesthetized with isoflurane were then transcardially perfused with pH 7.4 NMDG artificial cerebrospinal fluid (aCSF), which contains (mM): NMDG 92, KCl 2.5, NaH$_2$PO$_4$ 1.25, NaHCO$_3$ 20, HEPES 10, glucose 25, Na-ascorbate 5, thiourea 2, Na-pyruvate 3, MgSO$_4$ 10, CaCl$_2$ 0.5, *N*-acetyl-L-cysteine 12. Once completion of perfusion, mice's brain was quickly and smoothly extracted and placed into cutting NMDG solution for frozen 30 s. And coronal sections at 270 μm were prepared with a vibratome (Campden 5100MZ-PLUS Vibrotome) and then transferred to a chamber with 34°C NMDG aCSF for the first 15 min of recovery. And subsequently, brain slices were further gently transferred into another chamber holding aCSF at room temperature containing HEPES, pH 7.4, which contains (mM): NaCl 92, KCl 2.5, NaH$_2$PO$_4$ 1.25, NaHCO$_3$ 20, HEPES 10, glucose 25, Na-ascorbate 5, thiourea 2, Na-pyruvate 3, MgSO$_4$ 10, CaCl$_2$ 0.5, *N*-acetyl-L-cysteine 12. After 40–60 min of recovery, the brain slices were placed into a recording chamber with normal aCSF, pH 7.4, which contains (mM): NaCl 125, KCl 2.5, glucose 11, NaHCO$_3$ 26, NaH$_2$PO$_4$ 1.25, CaCl$_2$ 2, and MgCl$_2$ 2 for neuronal recording.

### Functional connectivity

To record optically evoked postsynaptic currents, we used an internal solution containing (mM): K-gluconate 130, KCl 10, HEPES 10, EGTA 1, MgCl$_2$ 2, Na$_2$-ATP 2, Na$_3$-GTP 0.4. Light pulses at 470 nm were

delivered through a light stimulator (Polygon400 DSI-E-0470-0590-NK1 Dynamic Spatial Illuminator) to activate POA ChR2-EGFP-expressing terminals while patched pPVT neurons were held at –70 mV. The neuron with access resistance <20 MΩ and leak current <100 pA was included for data collection.

### Measurements of synaptic activity and intrinsic properties

To measure the changes in synaptic transmission of pPVT neurons after chronic heat exposure and acute heat exposure, coronal mice's brain slices were prepared as mentioned above. The internal solution for both miniature inhibitory (sIPSCs, held at +10 mV) and excitatory (sEPSCs, held at –70 mV) postsynaptic currents recording was cesium-gluconate-based solution with pH 7.3–7.4 which contains (mM): CsMSF 130, HEPES 10, EGTA 1, $MgCl_2$-$6H_2O$ 2, $Na_2ATP$ 2, NaGTP 0.4, QX-314-Cl 5, TEA-OH 5. And TTX (0.5 mM) was present for both mEPSCs and mIPSCs recording. Once completion of recording, brain slices were fixed overnight with 4% PFA and the correct position of recorded pPVT neurons was identified by staining with streptavidin-conjugated Alex Fluor 405 (Thermo Fisher). For measuring the firing frequency under current clamp recording, steady-state current was injected in +10 pA increments from 0 to 100 pA. All action potential properties and excitability recordings in *Figure 6H* were performed in the presence of 10 μM CNQX and 100 μM picrotoxin. As mentioned in the previous paper (*Knowland et al., 2017*), the firing rate evoked by current injections and the half-width were calculated in Clampfit software (Molecular Devices) by using peak detection function and full-width at half-max amplitude, respectively. The threshold was measured as the change in the voltage from rest at which the slope = 20 V/s. Membrane resistance was calculated from the change in voltage elicited after a 50 ms 5 mV hyperpolarizing step from –70 mV (the last 10 ms of the step from baseline was taken as △V). Capacitance and membrane time constant were calculated by Clampfit during the first minute after breaking into cell.

### Long-term potentiation

To measure the neuroplasticity of the POA to pPVT pathway, two batches of mice with enough expression of *Camk2a*-ChR2-mCherry within the POA were prepared: one serving as the control group and the other as the chronic heat group. Sagittal brain slices from the mice were prepared, and a low-chloride internal solution was used to record pPVT neurons. EPSCs evoked by focal optogenetic stimulation through the objective (approximately 2–3 mm above the surface of brain slice) in a top-down vertical direction were elicited using a Polygon400 Multiwavelength Dynamic Patterned Illuminator (Mightex). A circular area (roughly 500 μm diameter) covering the recorded cell soma was illuminated with brief pulses of blue light (470 nm, duration: 2 ms). LTP induced by optogenetic stimulation of pPVT neurons was elicited by HFS consisting of three episodes of 30 Hz blue light pulses at 20 s intervals.

All above signals were collected using a MultiClamp 700B amplifier controlled by Clampfix 10.4 software via a Digital data 1550 interface (Molecular Devices). Electrical signals were filtered at 3 kHz, digitized at 10 kHz, and further analyzed using Clampfit 10.7 (Molecular Devices).

## Statistical analysis

Statistical analysis was performed using GraphPad Prism 8.0. Values were shown as mean ± standard error of the mean (SEM). Student's t-test (paired or unpaired, parametric or nonparametric), one-way ANOVA with Tukey post hoc test, or two-way ANOVA followed by Sidak post hoc test, were conducted for statistical analysis. The value for statistical significance was $p < 0.05$. For each experiment, the detailed statistics are described in the corresponding figure legend.

## Acknowledgements

We thank Dr. Tao Xu, Romeo Goh, Yun Zhu, and Kanglin Rong for their technical assistance.

## Additional information

### Funding

| Funder | Grant reference number | Author |
|---|---|---|
| University Grants Committee | C4012-22G | Wing-Ho Yung |
| University Grants Committee | 09203236 | Ya Ke |
| University Grants Committee | C6034-21G | Wing-Ho Yung |

The funders had no role in study design, data collection and interpretation, or the decision to submit the work for publication.

### Author contributions

Zhiping Cao, Conceptualization, Data curation, Software, Formal analysis, Validation, Investigation, Methodology, Writing – original draft; Wing-Ho Yung, Conceptualization, Resources, Supervision, Funding acquisition, Visualization, Writing – review and editing; Ya Ke, Conceptualization, Resources, Supervision, Funding acquisition, Investigation, Writing – review and editing

### Author ORCIDs

Zhiping Cao (ID) https://orcid.org/0000-0001-5560-7385
Wing-Ho Yung (ID) https://orcid.org/0000-0002-5542-8173
Ya Ke (ID) https://orcid.org/0000-0001-8166-6653

### Ethics

All experimental procedures were approved by the Animal Experimentations and Ethics Committee at the Chinese University of Hong Kong. All surgeries were performed under deep surgical anesthesia and every effort was made to minimize animal suffering.

Reviewer #1 (Public review): https://doi.org/10.7554/eLife.101302.3.sa1
Reviewer #2 (Public review): https://doi.org/10.7554/eLife.101302.3.sa2
Reviewer #3 (Public review): https://doi.org/10.7554/eLife.101302.3.sa3
Author response https://doi.org/10.7554/eLife.101302.3.sa4

## Additional files

### Supplementary files

MDAR checklist
Source data 1. The raw data of *Figures 1–6* for statistical analysis.

### Data availability

The source data underlying Figs 1C, E, G-I, L, M-Q, 2H, K, 3D-H, 4C, E, G-I, L, M, P-W, 5B-E, H-L, O-S, V, 6C-F, H, I, L, M and Figure 1-figure supplement 1A-I, Figure 1-figure supplement 2B, D, F, G, H, K, L, Figure 2-figure supplement 1C, Figure 3-figure supplement 1C, D, Figure 3-figure supplement 2C, D, G, I, Figure 5-figure supplement 1C, F, G-J, M, Figure 6-figure supplement 1A-F are provided as a Source Data file.

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
