## [Editor Report · eLife Assessment]

This **important** study identifies one way in which episodic heat exposure can result in negative changes in motivated and affective behaviors. This work positively expands the field of thermoregulation. The data were collected using a myriad of next-generation approaches, including extensive behavior testing, thermal monitoring, electrophysiology, circuit mapping, and manipulations. There is **convincing** evidence that neurons of the paraventricular thalamus change plastically over three weeks of episodic heat stimulation this affects behavioral outputs such as social interactions and anxiety-related behavior. Conclusions regarding the specificity of the POA-pPVT pathway compared to other inputs to the PVT in the control of observed effects would benefit from further validation. The study will be of interest to behavioral neuroscientists, climate/environmental biologists, and pre-clinical neuropsychiatrists.

---

## [Referee Report · Reviewer #1 (Public review)]

Summary:

The manuscript by Cao et al. examines an important but understudied question of how chronic exposure to heat drives changes in affective and social behaviors. It has long been known that temperature can be a potent driver of behaviors and can lead to anxiety and aggression. However, the neural circuitry that mediates these changes is not known. Cao et al. take on this question by integrating optical tools of systems neuroscience to record and manipulate bulk activity in neural circuits, in combination with a creative battery of behavior assays. They demonstrate that chronic daily exposure to heat leads to changes in anxiety, locomotion, social approach, and aggression. They identify a circuit from preoptic area (POA) to posterior paraventricular thalamus (pPVT) in mediating these behavior changes. The POA-PVT circuit increases activity during heat exposure. Further, manipulation of this circuit can drive affective and social behavioral phenotypes even in the absence of heat exposure. Moreover, silencing this circuit during heat exposure prevents the development of negative phenotypes. Overall the manuscript makes an important contribution to the understudied area of how ambient temperature shapes motivated behaviors.

Strengths

The use of state-of-the-art systems neuroscience tools (in vivo optogenetics and fiber photometry, slice electrophysiology), chronic temperature-controlled experiments, and a rigorous battery of behavioral assays to determine affective phenotypes. The optogenetic gain of function of affective phenotypes in the absence of heat, and loss of function in the presence of heat are very convincing manipulation data. Overall a significant contribution to the circuit-level instantiation of temperature induced changes in motivated behavior, and creative experiments.

Weaknesses

The authors have fully addressed all of my questions and concerns, with the exception of one comment. They mention that they did carry out measurements of core body temperature as a control during optogenetic experiments and did not see any effects. However, I could only find this reported in the text but could not find the data in the main or supplementary figures.

---

## [Referee Report · Reviewer #2 (Public review)]

Summary:

The study by Cao et al. highlights an interesting and important aspect of heat- and thermal biology: the effect of repetitive, long-term heat exposure and its impact on brain function.

Even though peripheral, sensory temperature sensors and afferent neuronal pathways conveying acute temperature information to the CNS have been well established, it is largely unknown how persistent, long-term temperature stimuli interact with and shape CNS function, and how these thermally-induced CNS alterations modulate efferent pathways to change physiology and behavior. This study is therefore not only novel but, given global warming, also timely.

The authors provide compelling evidence that neurons of the paraventricular thalamus change plastically over three weeks of episodic heat stimulation and they convincingly show that these changes affect behavioral outputs such as social interactions, and anxiety related behaviors.

Strengths:

• It is impressive that the assessed behaviors can be (i) recruited by optogenetic fiber activation and (ii) inhibited by optogenetic fiber inhibition when mice are exposed to heat. Technically, when/how long is the fiber inhibition performed? It says in the text "3 min on and 3 min off". Is this only during the 20 minutes heat stimulation or also at other times?

• It is interesting that the frequency of activity in pPVT neurons, as assessed by fiber photometry, stays increased after long-term heat exposure (day 22) when mice are back at normal room temperature. This appears similar to a previous study that found long-term heat exposure to transform POA neurons plastically to become tonically active (https://www.biorxiv.org/content/10.1101/2024.08.06.606929v1). Interestingly, the POA neurons that become tonically active by persistent heat exposure described in the above study are largely excitatory and thus these could drive the activity of the pPVT neurons analyzed in this study.

How can it be reconciled that the majority of the inputs from the POA are found to be largely inhibitory (Fig. 2H)? Is it possible that this result stems from the fact that non-selective POA-to-pPVT projections are labelled by the approach used in this study and not only those pathways activated by heat? These points would be nice to discuss.

• It is very interesting that no LTP can be induced after chronic heat exposure (Fig. K-M); the authors suggest that "the pathway in these mice were already saturated" (line 375). Could this hypothesis be tested in slices by employing a protocol to extinguish pre-existing (chronic heat exposure-induced) LTP? This would provide further strength to the findings/suggestion that an important synaptic plasticity mechanism is at play that conveys behavioral changes upon chronic heat stimulation.

• It is interesting that long-term heat does not increase parameters associated with depression (Fig. 1N-Q), how is it with acute heat stress, are those depression parameters increased acutely? It would be interesting to learn if "depression indicators" increase acutely but then adapt (as a consequence of heat acclimation) or if they are not changed at all and are also low during acute heat exposure.

---

## [Referee Report · Reviewer #3 (Public review)]

In this study, Cao et al. explore the neural mechanisms by which chronic heat exposure induces negative valence and hyperarousal in mice, focusing on the role of the posterior paraventricular nucleus (pPVT) neurons that receive projections from the preoptic area (POA). The authors show that chronic heat exposure leads to heightened activity of the POA projection-receiving pPVT neurons, potentially contributing to behavioral changes such as increased anxiety level and reduced sociability, along with heightened startle responses. In addition, using electrophysiological methods, the authors suggest that increased membrane excitability of pPVT neurons may underlie these behavioral changes. The use of a variety of behavioral assays enhances the robustness of their claim. Moreover, while previous research on thermoregulation has predominantly focused on physiological responses to thermal stress, this study adds a unique and valuable perspective by exploring how thermal stress impacts affective states and behaviors, thereby broadening the field of thermoregulation.

While the manuscript has been revised and some efforts have been made to address the reviewers' concerns, the majority of the issues raised remain insufficiently resolved. Therefore, the reviewer has highlighted key major points that the authors should address to strengthen the manuscript's conclusions.

Major points

The manuscript highlights the increased activity in pPVT neurons receiving projections from the POA (Figure 3) and shows that these neurons are necessary for heat-induced behavioral changes (Figures 4N-W). However, it remains unclear whether the POA-to-pPVT projection itself plays a critical role. Since pPVT recipient neurons can receive inputs from various brain regions, the role of the POA input in driving these effects needs to be validated more explicitly.

(1) To establish this, the authors should conduct experiments directly inhibiting the POA-to-pPVT projection and demonstrate whether the increased activity in pPVT neurons due to chronic heat exposure is abolished when the POA is blocked.

(2) Alternatively, the authors could use anterograde labeling from the POA and specifically target recipient neurons in the pPVT to confirm that the observed excitatory inputs originate from the POA (related to Figure 6).

(3) If these experiments are not feasible, the authors should consider toning down the emphasis on the POA's role throughout the manuscript and discussing this limitation explicitly. The term "POA recipient pPVT neurons" should be used consistently to avoid misleading implications that the POA-to-pPVT excitatory projection is definitively established as the key pathway.

a) For example, in lines 368-369, the phrase "The increase in presynaptic excitability of the POA to pPVT excitatory pathway" represents a logical jump, as the data only support the "differential increase in presynaptic excitability of the excitatory pathway" (as described in lines 358-359) without specifically confirming the POA-to-pPVT pathway.

b) Similarly, in lines 442-446, the statement "the role of excitatory projections from POA to pPVT in chronic heat exposure-induced emotional changes" should be revised to "the role of excitatory projection recipient pPVT in chronic heat~," as the data do not provide direct evidence that heat-responsive POA neurons projecting to pPVT mediate these effects. Such revisions would improve clarity and ensure that the conclusions remain aligned with the presented data.

---

## [Author Response]

The following is the authors’ response to the original reviews.

**Reviewer #1 (Public review):**
Summary:The manuscript by Cao et al. examines an important but understudied question of how chronic exposure to heat drives changes in affective and social behaviors. It has long been known that temperature can be a potent driver of behaviors and can lead to anxiety and aggression. However, the neural circuitry that mediates these changes is not known. Cao et al. take on this question by integrating optical tools of systems neuroscience to record and manipulate bulk activity in neural circuits, in combination with a creative battery of behavior assays. They demonstrate that chronic daily exposure to heat leads to changes in anxiety, locomotion, social approach, and aggression. They identify a circuit from the preoptic area (POA) to the posterior paraventricular thalamus (pPVT) in mediating these behavior changes. The POA-PVT circuit increases activity during heat exposure. Further, manipulation of this circuit can drive affective and social behavioral phenotypes even in the absence of heat exposure. Moreover, silencing this circuit during heat exposure prevents the development of negative phenotypes. Overall the manuscript makes an important contribution to the understudied area of how ambient temperature shapes motivated behaviors.Strengths:The use of state-of-the-art systems neuroscience tools (in vivo optogenetics and fiber photometry, slice electrophysiology), chronic temperature-controlled experiments, and a rigorous battery of behavioral assays to determine affective phenotypes. The optogenetic gain of function of affective phenotypes in the absence of heat, and loss of function in the presence of heat are very convincing manipulation data. Overall a significant contribution to the circuit-level instantiation of temperature-induced changes in motivated behavior, and creative experiments.Weaknesses：(1) There is no quantification of cFos/rabies overlap shown in Figure 2, and no report of whether the POA-PVT circuit has a higher percentage of Fos+ cells than the general POA population. Similarly, there is no quantification of cFos in POA recipient PVT cells for Figure 2 Supplement 2.

Thanks for the comment. The quantification results of c-Fos signal have been provided in the main text and figures.

(2) The authors do not address whether stimulation of POA-PVT also increases core body temperature in Figure 3 or its relevant supplements. This seems like an important phenotype to make note of and could be addressed with a thermal camera or telemetry.

Thanks for raising this point. We did indeed monitor the core body temperature during stimulation of POA-PVT pathway, but we did not observe any significant changes. We have included this finding in the revised manuscript.

(3) In Figure 3G: is Day 1 vs Day 22 "pre-heat" significant? The statistics are not shown, but this would be the most conclusive comparison to show that POA-PVT cells develop persistent activity after chronic heat exposure, which is one of the main claims the authors make in the text. This analysis is necessary in order to make the claim of persistent circuit activity after chronic heat exposure.

Figure 3G does compare the Day 1 preheat to Day22 preheat, and the difference was significant. The wording has been corrected to avoid confusion. Also, we have modified Figure 3D to 3H in our revised manuscript to improve the clarity of these plots.

(4) In Figure 4, the control virus (AAV1-EYFP) is a different serotype and reporter than the ChR2 virus (AAV9-ChR2-mCherry). This discrepancy could lead to somewhat different baseline behaviors.

Thanks for bringing out this issue. We acknowledge that using AA1-EGFP (a different serotype and reporter compared to the AAV9-ChR2-mCherry) as our control virus is not ideal. But based on our own prior experiments, we observed no significant differences in baseline behaviors between animals injected with AAV1 and AAV9 EYFP as well as control mice without virus injection. Therefore, we believe that the baseline behaviors of the animals were unaffected.

(5) In Figure 5G, N for the photometry data: the authors assess the maximum z-score as a measure of the strength of calcium response, however the area under the curve (AUC) is a more robust and useful readout than the maximum z score for this. Maximum z-score can simply identify brief peaks in amplitude, but the overall area under the curve seems quite similar, especially for Figure 5N.

Thanks for the comment. We agree with the reviewer that the area under the curve (AUC) is an alternative readout for measurement of the strength of calcium response. However, the reason why we chose the maximum z-score is based on the observation that we found POA recipient pPVT neurons after chronic heat treatment exhibited a higher calcium peak corresponding to certain behavioral performances when compared to pre-heat conditions. We thus applied the maximum z-score as a representative way to describe the neuronal activity changes of mice during certain behaviors before and after chronic heat treatment. The other consideration is that we want to reflect that POA recipient pPVT neurons become more sensitive and easier to be activated after chronic heat exposure under the same stressful situations compared to control mice. The maximum z score represented by peak in combination with particular behavioral performances is considered more suitable to highlight our findings in this study.

(6) For Fig 5V: the authors run the statistics on behavior bouts pooled from many animals, but it is better to do this analysis as an animal average, not by compiling bouts. Compiling bouts over-inflates the power and can yield significant p values that would not exist if the analysis were carried out with each animal as an n of 1.

Thanks for the comment and suggestion. We had tried both methods and the statistical results were similar. As suggested, we have updated Fig 5V, as well as Fig. 5H and 5O by comparing animal average in our revised manuscript.

(7) In general this is an excellent analysis of circuit function but leaves out the question of whether there may be other inputs to pPVT that also mediate the same behavioral effect. Future experiments that use activity-dependent Fos-TRAP labeling in combination with rabies can identify other inputs to heat-sensitive pPVT cells, which may have convergent or divergent functions compared to the POA inputs.

Thanks for the valuable suggestion, which would enhance the conclusion. We will consider adopting this approach in future investigations into this question.

**Reviewer #2 (Public review):**
SummaryThe study by Cao et al. highlights an interesting and important aspect of heat- and thermal biology: the effect of repetitive, long-term heat exposure and its impact on brain function.Even though peripheral, sensory temperature sensors and afferent neuronal pathways conveying acute temperature information to the CNS have been well established, it is largely unknown how persistent, long-term temperature stimuli interact with and shape CNS function, and how these thermally-induced CNS alterations modulate efferent pathways to change physiology and behavior. This study is therefore not only novel but, given global warming, also timely.The authors provide compelling evidence that neurons of the paraventricular thalamus change plastically over three weeks of episodic heat stimulation and they convincingly show that these changes affect behavioral outputs such as social interactions, and anxiety-related behaviors.Strengths(1) It is impressive that the assessed behaviors can be (i) recruited by optogenetic fiber activation and (ii) inhibited by optogenetic fiber inhibition when mice are exposed to heat. Technically, when/how long is the fiber inhibition performed? It says in the text "3 min on and 3 min off". Is this only during the 20-minute heat stimulation or also at other times?

Thanks for pointing out the need for clarification. Our optogenetic inhibition had been conducted for 21 days during the heat exposure period (90 mins) for each mouse. And to avoid the light-induced heating effect, we applied the cyclical mode of 3 minutes’ light on and 3 minutes’ light off only during the process of heat exposure but not other time. The detailed description has been supplemented in the Method part of our revised manuscript.

(2) It is interesting that the frequency of activity in pPVT neurons, as assessed by fiber photometry, stays increased after long-term heat exposure (day 22) when mice are back at normal room temperature. This appears similar to a previous study that found long-term heat exposure to transform POA neurons plastically to become tonically active (https://www.biorxiv.org/content/10.1101/2024.08.06.606929v1). Interestingly, the POA neurons that become tonically active by persistent heat exposure described in the above study are largely excitatory, and thus these could drive the activity of the pPVT neurons analyzed in this study.

Thanks for pointing out this study that suggests similar plasticity of POA neurons under long-term heat exposure serving a different purpose. We have included this information in our discussion as well.

(3) How can it be reconciled that the majority of the inputs from the POA are found to be largely inhibitory (Fig. 2H)? Is it possible that this result stems from the fact that non-selective POA-to-pPVT projections are labelled by the approach used in this study and not only those pathways activated by heat? These points would be nice to discuss.

Thanks for raising these important questions. Although it is not our primary focus, we are aware of the substantial inhibitory inputs from POA to pPVT which suggests an important function. However, we do not think that this pathway, which would exert an opposite effect on POA-recipient pPVT neurons compared to the excitatory input, contributes to the long-term effect of chronic heat exposure. This is due to the increased, rather than decreased, excitability of the neurons. There is a possibility that this inhibitory input serves as a short-term inhibitory control for other purpose. Further work is needed to fully address this question.

(4) It is very interesting that no LTP can be induced after chronic heat exposure (Figures K-M); the authors suggest that "the pathway in these mice were already saturated" (line 375). Could this hypothesis be tested in slices by employing a protocol to extinguish pre-existing (chronic heat exposure-induced) LTP? This would provide further strength to the findings/suggestion that an important synaptic plasticity mechanism is at play that conveys behavioral changes upon chronic heat stimulation.

We agree with the reviewer that the results of the suggested experiment would further strengthen our hypothesis. We will try to confirm this in future studies.

(5) It is interesting that long-term heat does not increase parameters associated with depression (Figure 1N-Q), how is it with acute heat stress, are those depression parameters increased acutely? It would be interesting to learn if "depression indicators" increase acutely but then adapt (as a consequence of heat acclimation) or if they are not changed at all and are also low during acute heat exposure.

Based on our observations, we did not find increased depression parameters after acute heat stress in our experiments (data not shown), which was consistent with other two previous studies (Beas et al., 2018; Zhang et al., 2021). It appears that acute heat stress is more associated with anxiety-like behavior and may not be sufficient to induce depression-like phenotypes in rodents, aligning with our observation during experiments.

Beas BS, Wright BJ, Skirzewski M, Leng Y, Hyun JH, Koita O, Ringelberg N, Kwon HB, Buonanno A, Penzo MA (2018) The locus coeruleus drives disinhibition in the midline thalamus via a dopaminergic mechanism Nat Neurosci 21:963-973.

Zhang GW, Shen L, Tao C, Jung AH, Peng B, Li Z, Zhang LI, Whit Tao HZ (2021) Medial preoptic area antagonistically mediates stress-induced anxiety and parental behavior Nat Neurosci 24:516-528.

Weaknesses/suggestions for improvement.(1) The introduction and general tenet of the study is, to us, a bit too one-sided/biased: generally, repetitive heat exposure --heat acclimation-- paradigms are known to not only be detrimental to animals and humans but also convey beneficial effects in allowing the animals and humans to gain heat tolerance (by strengthening the cardiovascular system, reducing energy metabolism and weight, etc.).

Thanks for the suggestion. We have modified the introduction in our revised manuscript to make it more balanced.

(2) The point is well taken that these authors here want to correlate their model (90 minutes of heat exposure per day) to heat waves. Nevertheless, and to more fully appreciate the entire biology of repetitive/chronic/persistent heat exposure (heat acclimation), it would be helpful to the general readership if the authors would also include these other aspects in their introduction (and/or discussion) and compare their 90-minute heat exposure paradigm to other heat acclimation paradigms. For example, many past studies (using mice or rats)m have used more subtle temperatures but permanently (and not only for 90 minutes) stimulated them over several days and weeks (for example see PMID: 35413138). This can have several beneficial effects related to cardiovascular fitness, energy metabolism, and other aspects. In this regard: 38{degree sign}C used in this study is a very high temperature for mice, in particular when they are placed there without acclimating slowly to this temperature but are directly placed there from normal ambient temperatures (22{degree sign}C-24{degree sign}C) which is cold/coolish for mice. Since the accuracy of temperature measurement is given as +/- 2{degree sign}C, it could also be 40{degree sign}C -- this temperature, 40{degree sign}C, non-heat acclimated C57bl/6 mice will not survive for long.The authors could consider discussing that this very strong, short episodic heat-stress model used here in this study may emphasize detrimental effects of heat, while more subtle long-term persistent exposure may be able to make animals adapt to heat, become more tolerant, and perhaps even prevent the detrimental cognitive effects observed in this study (which would be interesting to assess in a follow-up study).

Thanks for pointing out the important aspect regarding the different heat exposure paradigms and their potential impacts. We have incorporated these points into both the Introduction and Discussion sections of the revised manuscript.

(3) Line 140: It would help to be clear in the text that the behaviors are measured 1 day after the acute heat exposure - this is mentioned in the legend to the figure, but we believe it is important to stress this point also in the text. Similarly, this is also relevant for chronic heat stimulation: it needs to be made very clear that the behavior is measured 1 day after the last heat stimulus. If the behaviors had been measured during the heat stimulus, the results would likely be very different.

Thanks for the suggestion, and we have clarified the procedure in the revised manuscript.

(4) Figure 2 D and Figure 2- Figure Supplement 1: since there is quite some baseline cFos activity in the pPVT region we believe it is important to include some control (room temperature) mice with anterograde labelling; in our view, it is difficult/not possible to conclude, based on Fig 2 supplement 2C, that nearly 100% of the cfos positive cells are contacted by POA fibre terminals (line 168). By eye there are several green cells that don't have any red label on (or next to) them; additionally, even if there is a little bit of red signal next to a green cell: this is not definitive proof that this is a synaptic contact. It is therefore advisable to revisit the quantification and also revisit the interpretation/wording about synaptic contacts.In relation to the above: Figure 2h suggests that all neurons are connected (the majority receiving inhibitory inputs), is this really the case, is there not a single neuron out of the 63 recorded pPVT neurons that does not receive direct synaptic input from the POA?

Thanks for the comments. For Figure 2-figure supplement 1, the baseline c-Fos activity in pPVT were indeed measured from mouse under room temperature. Observed activity may be attributed to the diverse functions that the pPVT is responsible for. Compared to the heat-exposed group, we observed significant increases in c-Fos signals, suggesting the effect of heat exposure.

For Figure 2-figure supplement 2, through targeted injection of AAV1-Cre into the POA, we achieved selective expression of Cre-dependent ChR2-mCherry in pPVT neurons receiving POA inputs. Following heat exposure, we observed substantial colocalization between heat-induced c-Fos expression (green signal) and ChR2-mCherry-labeled neurons (red signal) in the pPVT. This extensive overlap indicates that POA-recipient pPVT neurons are predominantly heat-responsive and likely mediate the behavioral alterations induced by chronic heat exposure. We have validated these signals and included updated quantification in our revised manuscript.

For Fig 2H, we specifically patched those neurons that were surrounded by red fluorescence under the microscope, ensuring that the patched neurons had a high likelihood of being innervated from POA. This is why all 63 recorded pPVT neurons were found to receive direct synaptic input from the POA.

(5) It would be nice to characterize the POA population that connects to the pPVT, it is possible/likely that not only warm-responsive POA neurons connect to that region but also others. The current POA-to-pPVT optogenetic fibre stimulations (Figure 4) are not selective for preoptic warm responsive neurons; since the POA subserves many different functions, this optogenetic strategy will likely activate other pathways. The referees acknowledge that molecular analysis of the POA population would be a major undertaking. Instead, this could be acknowledged in the discussion, for example in a section like "limitation of this study".

Thanks for the suggestion. We have supplemented this part in our revised manuscript.

(6) Figure 3a the strategy to express Gcamp in a Cre-dependent manner: it seems that the Gcamp8f signal would be polluted by EGFP (coming from the Cre virus injected into the POA): The excitation peak for both is close to 490nm and emission spectra/peaks of GCaMP8f (510-520 nm) and EGFP (507-510 nm) are also highly overlapping. We presume that the high background (EGFP) fluorescence signal would preclude sensitive calcium detection via Gcamp8f, how did the authors tackle this problem?

Thank you for pointing out this issue. We acknowledge that we included AAV1-EGFP when recording the GCaMP8F signal to assist in the post-verification of the accuracy of the injection site. But we also collected recording data from mice with AAV1-Cre without EGFP injected into POA and Cre-dependent GCaMP8F in pPVT, albert in a smaller number. We did not observe any obvious differences in the change in calcium signal between these two virus strategies, suggesting that the sensitivity of the GCaMP signals was not significantly affected by the increased baseline fluorescence due to EGFP.

(7) How did the authors perform the social interaction test (Figures 1F, G)? Was the intruder mouse male or female? If it was a male mouse would the interaction with the female mouse be a form of mating behavior? If so, the interpretation of the results (Figures 1F, G) could be "episodic heat exposure over the course of 3 weeks reduces mating behavior".

Thanks for the comment. For this female encounter test, we strictly followed the protocol by Ago Y, et al., (2015). During this test, both the strange male and female mice were placed into a wired cup (which is made up of mental wire entanglement and the size for each hole is 0.5 cm [L] x 0.5 cm [W]), which successfully prevented large body contact and the mating behavior but only innate sex-motivated moving around the cup. We have supplemented the details in the method part of our revised manuscript.

Ago Y, Hasebe S, Nishiyama S, Oka S, Onaka Y, Hashimoto H, Takuma K, Matsuda T (2015) The Female Encounter Test: A Novel Method for Evaluating Reward-Seeking Behavior or Motivation in Mice Int J Neuropsychopharmacol 18: pyv062.

**Reviewer #3 (Public review):**
In this study, Cao et al. explore the neural mechanisms by which chronic heat exposure induces negative valence and hyperarousal in mice, focusing on the role of the posterior paraventricular nucleus (pPVT) neurons that receive projections from the preoptic area (POA). The authors show that chronic heat exposure leads to heightened activity of the POA projection-receiving pPVT neurons, potentially contributing to behavioral changes such as increased anxiety level and reduced sociability, along with heightened startle responses. In addition, using electrophysiological methods, the authors suggest that increased membrane excitability of pPVT neurons may underlie these behavioral changes. The use of a variety of behavioral assays enhances the robustness of their claim. Moreover, while previous research on thermoregulation has predominantly focused on physiological responses to thermal stress, this study adds a unique and valuable perspective by exploring how thermal stress impacts affective states and behaviors, thereby broadening the field of thermoregulation. However, a few points warrant further consideration to enhance the clarity and impact of the findings.(1) The authors claim that behavior changes induced by chronic heat exposure are mediated by the POA-pPVT circuit. However, it remains unclear whether these changes are unique to heat exposure or if this circuit represents a more general response to chronic stress. It would be valuable to include control experiments with other forms of chronic stress, such as chronic pain, social defeat, or restraint stress, to determine if the observed changes in the POA-pPVT circuit are indeed specific to thermal stress or indicative of a more universal stress response mechanism.

We also share similar considerations as the reviewer and indeed have conducted experiments to explore this possibility. Our findings suggest that the POA-pPVT pathway may also mediate behavioral changes induced by other chronic stress, e.g. chronic restraint stress. Nevertheless, given the well-known prominent role of POA neurons in heat perception, we do believe that the POA-pPVT has a specialized role in mediating chronic heat induced changes. The role of this pathway in other stress-related responses will need a more comprehensive study in the future.

(2) The authors use the term "negative emotion and hyperarousal" to interpret behavioral changes induced by chronic heat (consistently throughout the manuscript, including the title and lines 33-34). However, the term "emotion" is broad and inherently difficult to quantify, as it encompasses various factors, including both valence and arousal (Tye, 2018; Barrett, L. F. 1999; Schachter, S. 1962). Therefore, the reviewer suggests the authors use a more precise term to describe these behaviors, such as valence. Additionally, in lines 117 and 137-139, replacing "emotion" with "stress responses," a term that aligns more closely with the physiological observations, would provide greater specificity and clarity in interpreting the findings.

Thanks for the suggestion. We have modified the description of “emotion” to “emotional valence” in various places throughout the revised manuscript.

(3) Related to the role of POA input to pPVT,a) The authors showed increased activity in pPVT neurons that receive projections from the POA (Figure 3), and these neurons are necessary for heat-induced behavioral changes (Figures 4N-W). However, is the POA input to the pPVT circuit truly critical? Since recipient pPVT neurons can receive inputs from various brain regions, the reviewer suggests that experiments directly inhibiting the POA-to-pPVT projection itself are needed to confirm the role of POA input. Alternatively, the authors could show that the increased activity of pPVT neurons due to chronic heat exposure is not observed when the POA is blocked. If these experiments are not feasible, the reviewer suggests that the authors consider toning down the emphasis on the role of the POA throughout the manuscript and discuss this as a limitation.b) In the electrophysiology experiments shown in Figures 6A-I, the authors conducted in vitro slice recordings on pPVT neurons. However, the interpretation of these results (e.g., "The increase in presynaptic excitability of the POA to pPVT excitatory pathway suggested plastic changes induced by the chronic heat treatment.", lines 349-350) appears to be an overclaim. It is difficult to conclude that the increased excitability of pPVT neurons due to heat exposure is specifically caused by inputs from the POA. To clarify this, the reviewer suggests the authors conduct experiments targeting recipient neurons in the pPVT, with anterograde labeling from the POA to validate the source of excitatory inputs.

For point (a), we acknowledge that pPVT neurons receiving POA inputs may also receive projections from other brain regions. While these additional inputs warrant investigation, they fall beyond the scope of our current study and represent promising directions for future research. Notably, compared to other well-characterized regions such as the amygdala and ventral hippocampus, the pPVT receives particularly robust projections from hypothalamic nuclei (Beas et al., 2018). Our optogenetic inhibition of POA-recipient pPVT neurons during chronic heat exposure effectively prevented the influence of POA excitatory projections on pPVT neurons. Furthermore, selective optogenetic activation of POA excitatory terminals within the pPVT was sufficient to induce similar behavioral abnormalities in mice, strongly supporting the causal role of POA inputs in mediating chronic heat exposure-induced behavioral alterations.

Beas BS, Wright BJ, Skirzewski M, Leng Y, Hyun JH, Koita O, Ringelberg N, Kwon HB, Buonanno A, Penzo MA (2018) The locus coeruleus drives disinhibition in the midline thalamus via a dopaminergic mechanism Nat Neurosci 21:963-973.

Regarding point (b), we acknowledge certain limitations in our in vitro patch-clamp recordings when attributing increased pPVT neuronal excitability to enhanced presynaptic POA inputs. Nevertheless, our brain slice recordings clearly demonstrated heightened excitability of pPVT neurons following chronic heat exposure. This finding was further corroborated by our in vivo fiber photometry recordings specifically targeting POA-recipient pPVT neurons, which confirmed that the increased pPVT neuronal activity was indeed modulated by POA inputs. The causal relationship was strengthened by our observation that optogenetic activation of POA excitatory terminals within the pPVT reproduced behavioral abnormalities similar to those observed in chronic heat-exposed mice. Additionally, our inability to induce circuit-specific LTP in the POA-pPVT pathway suggests that these synapses were already potentiated and saturated, reflecting enhanced excitatory inputs from the POA to pPVT. Collectively, these findings support our conclusion that increased excitatory projections from the POA to pPVT likely represent a key mechanism underlying chronic heat exposure-induced behavioral alterations in mice.

(4) The authors focus on the excitatory connection between the POA and pPVT (e.g., "Together, our results indicate that most of the pPVT-projecting POA neurons responded to heat treatment, which would then recruit their downstream neurons in the pPVT by exerting a net excitatory influence.", lines 169-171). However, are the POA neurons projecting to the pPVT indeed excitatory? This is surprising, considering (i) the electrophysiological data shown in Figures 2E-K that inhibitory current was recorded in 52.4% of pPVT neurons by stimulation of POA terminal, and (ii) POA projection neurons involved in modulating thermoregulatory responses to other brain regions are primarily GABAergic (Tan et al., 2016; Morrison and Nakamura, 2019). The reviewer suggests showing whether the heat-responsive POA neurons projecting to the pPVT are indeed excitatory (This could be achieved by retrogradely labeling POA neurons that project to the pPVT and conducting fluorescence in situ hybridization (FISH) assays against Slc32a1, Slc17a6, and Fos to label neurons activated by warmth). Alternatively, demonstrate, at least, that pPVT-projecting POA neurons are a distinct population from the GABAergic POA neurons that project to thermoregulatory regions such as DMH or rRPa. This would clarify how the POA-pPVT circuit integrates with the previously established thermoregulatory pathways.

Thanks for the comment and suggestion. We acknowledge that there are both excitatory and inhibitory projections from POA to pPVT. Although it is not our primary focus, we are aware of the substantial inhibitory inputs from POA to pPVT which suggests an important function. However, we do not think that this pathway, which would exert an opposite effect on POA-recipient pPVT neurons compared to the excitatory input, contributes to the long-term effect of chronic heat exposure. This is due to the increased, rather than decreased, excitability of the neurons. There is a possibility that this inhibitory input serves as a short-term inhibitory control for other purpose. Further work is needed to fully address this question.

**Recommendations for the authors:**

**Reviewer #1 (Recommendations for the authors):**
I have a number of suggested minor edits that would improve the readability and interpretation of figures for the reader. In many figures, there are places where it is unclear what is being tested, and making minor changes would make the manuscript flow more easily for the reader:(1) The authors could add additional details about the behavior paradigms in the Figures, especially Figure 1. How long was the chronic heat exposure for? At what temperature? What is the length of time between the end of heat exposure and the start of behaviors? What was the schedule of testing for EPM and social behaviors? Was it all on the same day or on different days? These details will make it easier for the reader to understand the behavior tests.

We have revised our experimental scheme, especially Figure 1, and added more detailed descriptions in the method section. The modifications have also been applied to the other figures.

(2) In Figures 1J and 1K, it is a bit unclear what is being shown in the right panel, since there are no axes or labels to interpret what is being plotted.

We have added body kinetics (purple dot) in the left panel of Figure 1J and 1K to align with the right panels, and we have updated our descriptions in the figure legend.

(3) In general, Figure 1 would benefit from more headers/labels or schematics to demonstrate what is being tested for example, it's unclear that forced swim, tail suspension, open field, aggression, sucrose preference, or acoustic startle are being studied unless the reader looks at the figure legend in depth. Simple schematics or titles for each panel would help.

We have added the abbreviated titles for each panel of Figure 1 to help readers to better understand what was being tested.

(4) Figure 2A would benefit from edits to the schematic so that it is clear that heat exposure is being done before the animal is sacrificed and cFos is stained.

We have revised the text to clarify that heat exposure occurred before the animal was sacrificed and c-Fos was stained.

(5) Figure 2D: would help if the quantification of overlap of cFos and rabies was shown in the figure in addition to reporting it in the text (84%).

We have added quantification in Figure 2D.

(6) The supplemental data in Figure 2 - Supplemental Figure 1 showing increased Fos in PVT and POA after heat exposure would actually help if it was in main Figure 2 so that the reader can more clearly see the rationale for choosing the POA-PVT circuit. But this is a matter of preference and up to the author where they want to show this data.

Thanks for the suggestion. But considering the layout and space, we will prefer to retain this part in Figure 2-supplemental figure 1.

(7) Figure 3 would benefit from a behavior schematic illustrating the time course of the experiment and what the heat exposure protocol is for each day (how many minutes heat 'on' vs 'off', the temperature of heat, etc). Also, what is different about day 22 that makes it chronic heat vs day 21? Currently, it is a bit hard to understand the protocol.

We have added the temperature and time of chronic heat exposure in the schematic of Figure 3. The “day 22” represented the time point after chronic heat exposure. And we measured the calcium activity of POA recipient pPVT neurons on day 22 to compare with day 1 to demonstrate that the activity changes of POA recipient pPVT neurons after chronic heat exposure.

(8) Figure 3D, it is unclear what the difference is between the Day 1 data on the left and Day 1 data on the right. Same with Figure 3H, unclear what the difference is between the left and the right.

The left panel and right panel reflect different parameters: frequency /min (left) and amplitude (△F/F) for Figure 3D-3H. By doing this, we want to reflect the dynamic activity changes of POA recipient pPVT neurons throughout chronic heat exposure process. Now, all figures in panel 3D to 3H have been revised to make them clearer in meaning.

(9) Figure 4A would benefit from schematics showing the stimulation protocol for chronic optogenetics (how many days? Frequency? Duration of time? Etc)

We have added detailed schematics in our Figure 4A.

**Reviewer #2 (Recommendations for the authors)**
(1) It is interesting that social behavior appears to be reduced upon long-term heat exposure but not after acute heat exposure. Interaction of animals, such as huddling, can be used by animals as a form of behavioral thermoregulation in cold environments and heat may drive animals apart to allow for better heat dissipation. The social interaction measured here is not huddling (because, I assume, the animals are separated by a divider?) but is this form of behavior measured here related to huddling/"social thermoregulation"? This could be discussed.

Our behavioral tests were performed at room temperature. Even though huddling is a type of social behavior, based on our observation, the tested mouse was actively revolving around the mental cap, suggesting this type of behavior is not related to huddling/social thermoregulation type of social behavior.

(2) Line 113: The statement "Chronic treatment did not change body temperature" should be clarified/rephrased because 90 minutes of 38 degrees centigrade exposure to heat will increase the body temperature of mice. It would be helpful if the authors made clear that they measure body temperature before the heat stimulus (and not during the heat stimulus), which is now only obvious if one digs into the methods section.

We have revised the text and clarified that body temperature was measured before the heat stimulus in the revised manuscript.

(3) Figure 1J and K: for the non-experts, these graphs are difficult to interpret, some more explanation is needed (what exactly is measured ?). We believe that the term "arousal" may not be justified in this context because the authors have not measured sleep patterns (EEG and EMG) to show that the mice arouse from a sleep (or sleep-like) stage; the authors may consider changing the terminology, e.g. something along the lines of "agitation" or "activity".

We have further elaborated the meaning of Figure 1J and K in our revised manuscript. The acoustic startle response is a well-recognized behavioral parameter reflecting arousal levels in rodent model. The more agitation in response to stimulus, the higher the arousal levels in mice. We have used the term “agitation” to describe mice’s performance in the acoustic startle response test.

**Reviewer #3 (Recommendations for the authors):**
(1) The authors suggest in the introduction of the manuscript that the HPA axis and other multifaceted factors may influence emotional changes caused by heat stress (lines 63-78). However, there are no experiments or discussions on how the POA-pPVT circuit interacts with these factors. In line with the study's proposed direction in the introduction section, it would be valuable to explore, or at least discuss, whether and how the POA-pPVT circuit interacts with the HPA axis or other neural circuits known to regulate emotional and stress responses. Alternatively, the reviewer suggests revising the content of the introduction to align with the focus of the study.

Although POA is known to possibly interact with the HPA axis via its connection with the paraventricular nucleus of the hypothalamus, there is hardly any evidence for the pPVT. Thus, we prefer not to speculate this question, which remains open, in our current manuscript.

(2) In Figure 5, the authors report that pPVT neurons that receive projections from the POA exhibited increased responses to stressful situations following chronic heat exposure. However, considering the long pre- and post-recording time gap of approximately three weeks, the additional expression of GCaMP protein over time could potentially account for the increased signal. Therefore, the reviewer recommends including a control group without heat exposure to rule out this possibility.

We have included Figure 3-figure supplement 1 in our manuscript to exclude the effect of expression of GCaMP protein over time on the recording of calcium signal.

(3) Related to Figure 2, (a) Please include quantification data of the overlap between retrogradely labeled and c-Fos-expressing POA neurons, which can be presented as a bar graph in Figure 2. This would be beneficial for readers to estimate how many warm-activated POA neurons connected to the pPVT are actively engaged under these conditions.

In the revised manuscript, we have included the quantification analysis in Figure 2.

b) The images in Figure 2 - Figure Supplement 1 seem to degrade in quality when magnified, making it difficult to discern finer details. Higher-resolution images would greatly improve the clarity and help in accurately visualizing the c-Fos expression patterns in the POA and pPVT regions.

We have changed our images of Figure 2-figure supplement 1 to higher-resolution in the revised manuscript.

c) The c-Fos images in Figure 2D and Figure 2 - Figure Supplement 2C appear unusual in that the c-Fos signal seems to fill the entire cell, whereas c-Fos protein is localized to the nucleus. Could the authors clarify whether this image accurately represents c-Fos staining or if there might be an issue with the staining or imaging process?

We are confident that the green signals in both Figure 2D and Figure 2-figure supplement 2C, which did not occupy the whole cell body, have already accurately reflected the c-Fos and that they were nucleus staining. We have updated the amplified picture in Figure 2D.

d) In Supplemental Figure 2B, the square marking the region of interest should be clearly explained in the figure legend to ensure that readers can fully understand the context and focus of the image.

We have further modified our figure legend in Figure 2-figure supplement 1 in our revised manuscript.